# Functional IKK/NF-κB signaling in pancreatic stellate cells is essential to prevent autoimmune pancreatitis

Lap Kwan Chan [1,2,3,6], Miltiadis Tsesmelis [1], Melanie Gerstenlauer[1], Frank Leithäuser[4], Alexander Kleger [5], Lukas Daniel Frick[3], Harald Jacob Maier[1,7,8] & Thomas Wirth [1,8 ✉]

Pancreatic stellate cells (PSCs) are resident cells in the exocrine pancreas which contribute to pancreatic fibrogenesis and inflammation. Studies on NF-κB in pancreatitis so far focused mainly on the parenchymal and myeloid compartments. Here we show a protective immunomodulatory function of NF-κB in PSCs. Conditional deletion of NEMO (IKKγ) in PSCs leads to spontaneous pancreatitis with elevated circulating IgM, IgG and antinuclear autoantibodies (ANA) within 18 weeks. When further challenged with caerulein, NEMO$^{\Delta Col1a2}$ mice show an exacerbated autoimmune phenotype characterized by increased infiltration of eosinophils, B and T lymphocytes with reduced latency period. Transcriptomic profiling shows that NEMO$^{\Delta Col1a2}$ mice display molecular signatures resembling autoimmune pancreatitis patients. Mechanistically, we show that PSC$^{\Delta NEMO}$ cells produce high levels of CCL24 ex vivo which contributes to eosinophil recruitment, as neutralization with a CCL24 antibody abolishes the transwell migration of eosinophils. Our findings uncover an unexpected immunomodulatory role specifically of NF-κB in PSCs during pancreatitis.

[1] Institute of Physiological Chemistry, University of Ulm, 89081 Ulm, Germany. [2] Institute of Molecular Cancer Research, University of Zurich, 8057 Zurich, Switzerland. [3] Department of Pathology and Molecular Pathology, University Hospital of Zurich, 8091 Zurich, Switzerland. [4] Institute of Pathology, University Hospital of Ulm, 89081 Ulm, Germany. [5] Department of Internal Medicine I, University Hospital of Ulm, 89081 Ulm, Germany. [6] Present address: Department of Pathology and Molecular Pathology, University Hospital of Zurich, 8091 Zurich, Switzerland. [7] Present address: Novartis Pharma AG, 4056 Basel, Switzerland. [8] These authors contributed equally: Harald Jacob Maier, Thomas Wirth. ✉email: thomas.wirth@uni-ulm.de

Autoimmune pancreatitis (AIP) is a rare form of chronic pancreatitis with characteristic clinicopathological features. Type I AIP (lymphoplasmacytic sclerosing pancreatitis) describes the pancreatic manifestation of IgG4-related diseases (IgG4-RDs), a systemic disease characterized by an abundant infiltration of IgG4+ plasma cells which can affect multiple organs[1–3]. In contrast, type II AIP exclusively affects the pancreas and is termed idiopathic duct-centric pancreatitis (IDCP) based on a characteristic infiltration pattern of neutrophil granulocytes to small pancreatic ducts known as granulocytic epithelial lesion (GEL). Though the etiology of AIP is still incompletely understood, different findings have suggested that genetic predisposing factors, immunologic triggers, microbiota and environmental factors can be involved in the pathogenesis of the disease[4]. Due to the complex immunologic interactions in AIP, current animal models of AIP only cover a limited spectrum of the disease. Therefore, new mouse models of AIP are highly desirable for a better understanding of the immunologic triggers and the pathogenetic mechanisms.

Nuclear factor κB (NF-κB) signaling has diverse functions in inflammation, cell survival and proliferation. Activation of the canonical NF-κB pathway requires the activity of the upstream IκB kinase (IKK) complex which is comprised of the IKK1 (IKKα) and IKK2 (IKKβ) catalytic subunits, as well as the regulatory subunit NEMO (IKKγ)[5]. NEMO plays an essential role in this activation process as its deletion renders cells unresponsive to stimuli of this pathway[6]. Several studies have highlighted the importance of IKK/NF-κB in acute and chronic pancreatitis. While blocking of canonical NF-κB in acinar cells can have a detrimental effect in pancreatitis, overactivation of this pathway can also trigger the disease[7–12]. Interestingly, ablation of IKK1 in acinar cells, although by a mechanism independent of its function in NF-κB signaling, also causes spontaneous pancreatitis[13]. In the endocrine pancreas, NF-κB inhibition in β cells promotes the development of autoimmune diabetes in nonobese diabetic mice[14]. In Pdx1 haploinsufficient mice, NF-κB inhibition promotes the disease progression into severe diabetes and the disease can be reverted by restoring the NF-κB activity in β cells[15].

Pancreatic stellate cells (PSCs) are resident cells in the exocrine pancreas which possess myofibroblast-like properties. So far, studies on NF-κB in pancreatitis focused mainly on the parenchymal and myeloid compartments[7–12,14]. Its function in PSCs during chronic pancreatitis has not been investigated. As one of the main players in pancreatic inflammation and fibrosis, activated PSCs are capable of producing multiple pro-inflammatory (e.g. TNF-α, RANTES, MCP-1, IL-1β and IL-6) and pro-fibrotic (e.g. TGF-β and CTGF) mediators, with most of these being target genes of NF-κB[16]. Therefore, inhibiting the NF-κB signaling in these cells may imply an appealing approach as a counter-inflammatory/fibrotic measure. Surprisingly, we found in this study that an inhibition of IKK/NF-κB in PSCs causes spontaneous inflammation in the pancreas. Caerulein treatment intensifies the phenotype leading to strong pancreatitis reminiscent of autoimmune pancreatitis. Isolated PSC$^{\Delta NEMO}$ cells express high levels of several chemokines including CCL24 which contributes to tissue eosinophilia. Our findings suggest a protective role of NF-κB in PSCs particularly in the immunomodulation process during pancreatitis.

## Results

### Long-term tamoxifen treatment leads to spontaneous pancreatitis in NEMO$^{\Delta Col1a2}$ mice.
To investigate the role of NF-κB in PSCs, we made use of the tamoxifen-inducible Col1a2-Cre.ERT construct to achieve a targeted deletion of *Ikbkg* (gene encoding NEMO) in the PSC population in adult mice (NEMO$^{\Delta Col1a2}$). Firstly, Cre recombinase activity was visualized in the pancreas using the NEMO$^{WT.RFP}$ reporter mice after receiving five daily tamoxifen injections, with RFP expression frequently observed but restricted to non-parenchymal cells (Supplementary Fig. 1a, b). Injection of tamoxifen without further exogenous insult did not result in any immediate pathological changes in NEMO$^{WT}$ and NEMO$^{\Delta Col1a2}$ animals after 3 weeks (Supplementary Fig. 1c). To examine the effect of long-term NEMO deletion in PSCs, mice from an age of 6–8 weeks were fed with tamoxifen food (400 mg/kg) for a period of 18 weeks (Fig. 1a). As reported previously[17], a change from chow to tamoxifen diet caused a drop in body weight (~5%) within the first week in both groups (Fig. 1b). However, NEMO$^{WT}$ animals quickly regained their body weight within the following 4 weeks while NEMO$^{\Delta Col1a2}$ animals experienced a further drop throughout the treatment period. Unexpectedly, NEMO$^{\Delta Col1a2}$ animals developed spontaneous inflammation in their pancreata with an increase in infiltration of macrophages (CD45 and F4/80 stainings) (Fig. 1c and Supplementary Fig. 2a). However, only animals showing strong pancreatic inflammation had elevated serum amylase levels (Supplementary Fig. 2b). Staining of the activation marker of PSCs (αSMA+) indicated an increase in cells adopting a myofibroblast phenotype (Fig. 1d). Interestingly, circulating IgG and IgM levels were significantly elevated in the serum of NEMO$^{\Delta Col1a2}$ animals (Fig. 1e). In parallel to an increase in total IgG and IgM, the level of antinuclear autoantibodies (ANA) was also significantly increased (Fig. 1f). These observed changes in the serological markers prompted us to hypothesize that the observed spontaneous pancreatitis phenotype in NEMO$^{\Delta Col1a2}$ animals represents an autoimmune disease. As autoimmune pancreatitis (AIP) can be the pancreatic manifestation of a systemic autoimmune disorder, we first screened through several organs like lung, kidney, intestine, spleen and skin of both NEMO$^{WT}$ and NEMO$^{\Delta Col1a2}$ animals but found no abnormalities (Supplementary Fig. 2c). In liver however, we found that 50% of the NEMO$^{\Delta Col1a2}$ animals exhibited hepatic inflammation with granulocytic infiltration (Supplementary Fig. 2d). The overall penetrance observed in NEMO$^{\Delta Col1a2}$ animals appeared to be consistent with the cross-organ spanning incidence as reported in AIP[4].

### Caerulein treatment induces severe inflammation with characteristics of autoimmune pancreatitis.
We then investigated whether a secretagogue-induced episode of chronic pancreatitis would affect the development and progression of the disease. We had previously shown that repetitive episodes of caerulein injection for two weeks results in a moderate chronic inflammation[7]. We used this protocol to treat both NEMO$^{WT}$ and NEMO$^{\Delta Col1a2}$ mice, accompanied with tamoxifen during the first week (Fig. 2a). Mice were either analyzed directly after caerulein treatment or one week later. Caerulein treatment resulted in a drop of pancreas weight which was comparable between NEMO$^{WT}$ and NEMO$^{\Delta Col1a2}$ mice at both time points (Supplementary Fig. 3a, b). Similarly, no significant difference was found in serum amylase and lipase activities at the 3-week time point, as these enzyme markers often reflect the acute phase of acinar cell damage (Supplementary Fig. 3c). Similar numbers of apoptotic acinar cells, as indicated by cleaved caspase 3, were observed in both groups (Supplementary Fig. 3d). However, administration of caerulein drastically accelerated the disease progression in NEMO$^{\Delta Col1a2}$ mice within 3 weeks, resulting in a significant reduction in the level of acinar cells and a massive infiltration of immune cells to the pancreas (Fig. 2b and Supplementary Fig. 3e, f). We therefore focused our characterization of the phenotype using the caerulein-treated groups. NEMO$^{WT}$ mice showed a

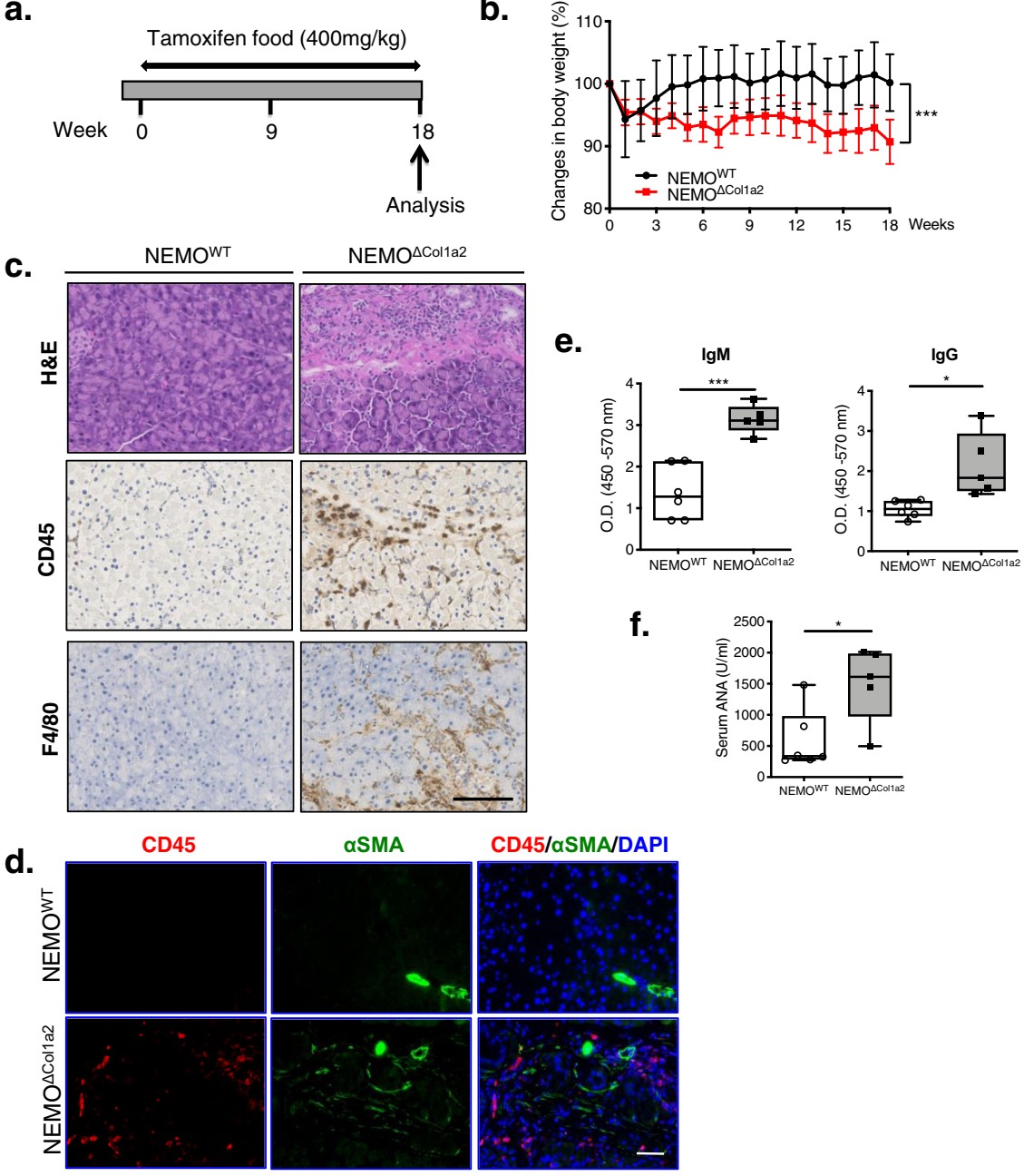

**Fig. 1 Spontaneous pancreatitis develops in NEMO$^{\Delta Col1a2}$ mice after long-term NEMO inhibition. a** Scheme of treatment with tamoxifen diet (400 mg/kg) for 18 weeks. **b** Body weight changes during the 18 weeks of treatment period. Mean±SD (NEMO$^{WT}$: $n = 9$; NEMO$^{\Delta Col1a2}$: $n = 8$). Area under the curve was compared. **c** Spontaneous pancreatitis and infiltration of immune cells observed in NEMO$^{\Delta Col1a2}$ pancreata. Scale bar: 100 μm. **d** Staining of CD45 and αSMA indicated an increase in activated pancreatic stellate cells at area where abundant immune cells were present. Scale bar: 50 μm. **e**, **f** Circulating IgM, IgG and ANA levels measured by ELISA on serum samples (NEMO$^{WT}$: $n = 6$; NEMO$^{\Delta Col1a2}$: $n = 5$). Whiskers: Min to Max. $T$-test (two-tailed): $*p < 0.05$; $***p < 0.001$. All n numbers represent biological replicates.

mild residual pancreatic inflammation and recovery was noticeable one week after the discontinuation of caerulein. In contrast, caerulein injections strengthened the features of autoimmunity in NEMO$^{\Delta Col1a2}$ mice characterized by persistent pancreatic inflammation with strong CD45$^+$ cell infiltration, marked fibrosis and sporadic venulitis (Fig. 2c, d). Particularly obliterative venulitis is pathogonomic for patients with AIP and has also been reported in other rodent models of autoimmune pancreatitis[18,19]. Metaplastic ductal lesions are frequently reported in pancreata bearing pancreatitis and pancreatic neoplasia. NEMO$^{\Delta Col1a2}$ mice showed more frequent acinar-to-ductal transdifferentiation in

their pancreata, which was characterized by the expression of both acinar and ductal markers (Fig. 2e). In line with this observation, re-activation of the transcription factor Sox9 in acinar cells was also frequently found in the NEMO$^{\Delta Col1a2}$ pancreata. Alcian blue staining indicated an increase in the frequency of mucinous metaplastic lesions in NEMO$^{\Delta Col1a2}$ animals (Fig. 2f). The observed transdifferentiation strongly associated with the severity of pancreatitis. We then asked the question if NEMO deletion contributed to extensive apoptosis in PSCs causing an exacerbated inflammation. Although there was a slight increase in the basal level of apoptotic PSCs (PDGFRβ$^+$/Cleaved

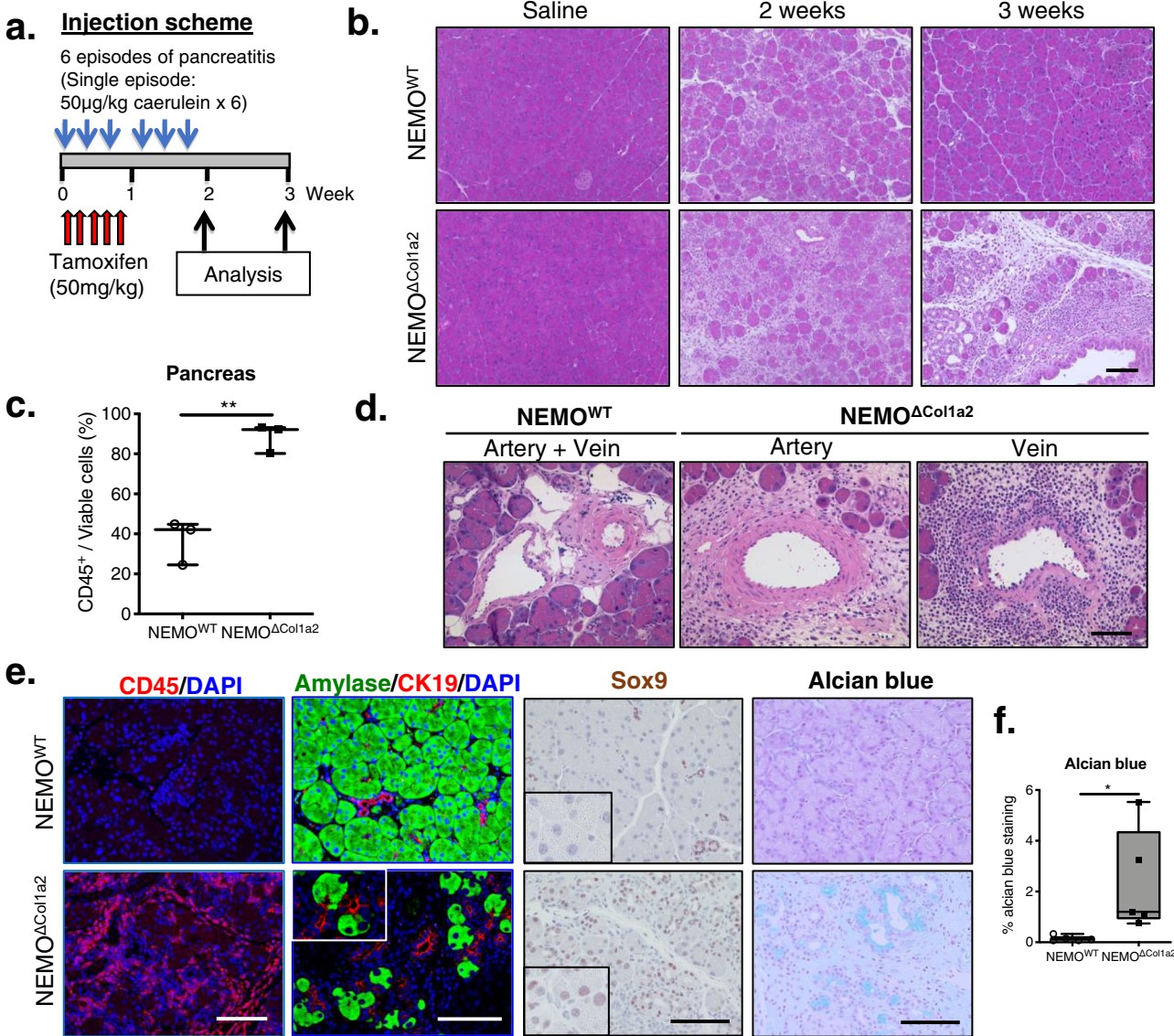

**Fig. 2 Deletion of NEMO in PSCs leads to exacerbated caerulein-induced pancreatitis. a** Injection scheme of tamoxifen (50 mg/kg daily for five days) and caerulein (6 episodes with each episode consisting of 6 hourly injections of 50 μg/kg of caerulein). **b** Pancreata from mice analyzed at the 2-week and 3-week time points. Scale bar: 100 μm. **c** Infiltrating cells from pancreata analyzed by FACS. The percentages of CD45$^+$/viable cells were measured ($n = 3$). **d** Sporadic venulitits found in NEMO$^{ΔCol1a2}$ pancreata. Scale bar: 50 μm. **e** Staining of pancreatic tissues (3 weeks) for infiltrating cells (CD45), markers of transdifferentiation (amylase and CK19; Sox9 in acinar cells) and mucin-producing ductal structures (alcian blue). DAPI: nuclei. Scale bar: 100 μm. **f** Quantification of alcian blue staining as a percentage of area ($n = 5$). Whiskers: Min to Max. T-test (two-tailed): *$p < 0.05$; **$p < 0.01$. All n numbers represent biological replicates.

Caspase 3$^+$) in the NEMO$^{ΔCol1a2}$ pancreata, the majority of PSCs appeared to be non-apoptotic (Supplementary Fig. 3d, g). Therefore, additional mechanisms are more likely to be involved in the exacerbated inflammatory phenotype.

**Deletion of NEMO in PSCs promotes tissue and peripheral eosinophilia.** Since we observed a massive infiltration of immune cells in the NEMO$^{ΔCol1a2}$ pancreata after caerulein treatment, we analyzed the contribution of different chemokine and chemokine receptor axes in the recruitment of immune cells. Quantitative PCR analysis revealed a strong upregulation of several chemokines (CCL6, CCL8, CCL9, CCL12 and CCL24) and chemokine receptors (CCR2, CCR3 and CXCR4) in the NEMO$^{ΔCol1a2}$ pancreata and among the most upregulated genes the CCL24/CCR3 axis (Fig. 3a). CCL24 (Eotaxin-2) is a potent chemoattractant which can act selectively on the receptor CCR3 to promote

the recruitment of eosinophils and play an important role in tissue eosinophilia[20]. To study the different immune cells, we performed cell isolation by gradient centrifugation from animals treated with caerulein and tamoxifen. Through FACS analysis, we observed an increase in the granulocyte population in NEMO$^{ΔCol1a2}$ animals (SSC-A vs. FSC-A), as well as Siglec-F$^+$ cells (Fig. 3b & Supplementary Fig. 4a). Co-staining of the CD45$^+$ population with additional myeloid markers also revealed the presence of the CD45$^+$CD11b$^+$CD11c$^-$Ly6G$^-$CCR3$^+$Siglec-F$^+$ eosinophil population, which was much more abundant in the NEMO$^{ΔCol1a2}$ group (23.7% vs. 0.4%) (Fig. 3c). Siglec-F is expressed primarily in eosinophils in mice but has been reported also in a subset of macrophages[21]. We therefore verified the result with a second eosinophil marker proteoglycan 2 (PRG2) which again showed a strong upregulation (Fig. 3d). Siglec-F immunofluorescence staining indicated an extensive infiltration of

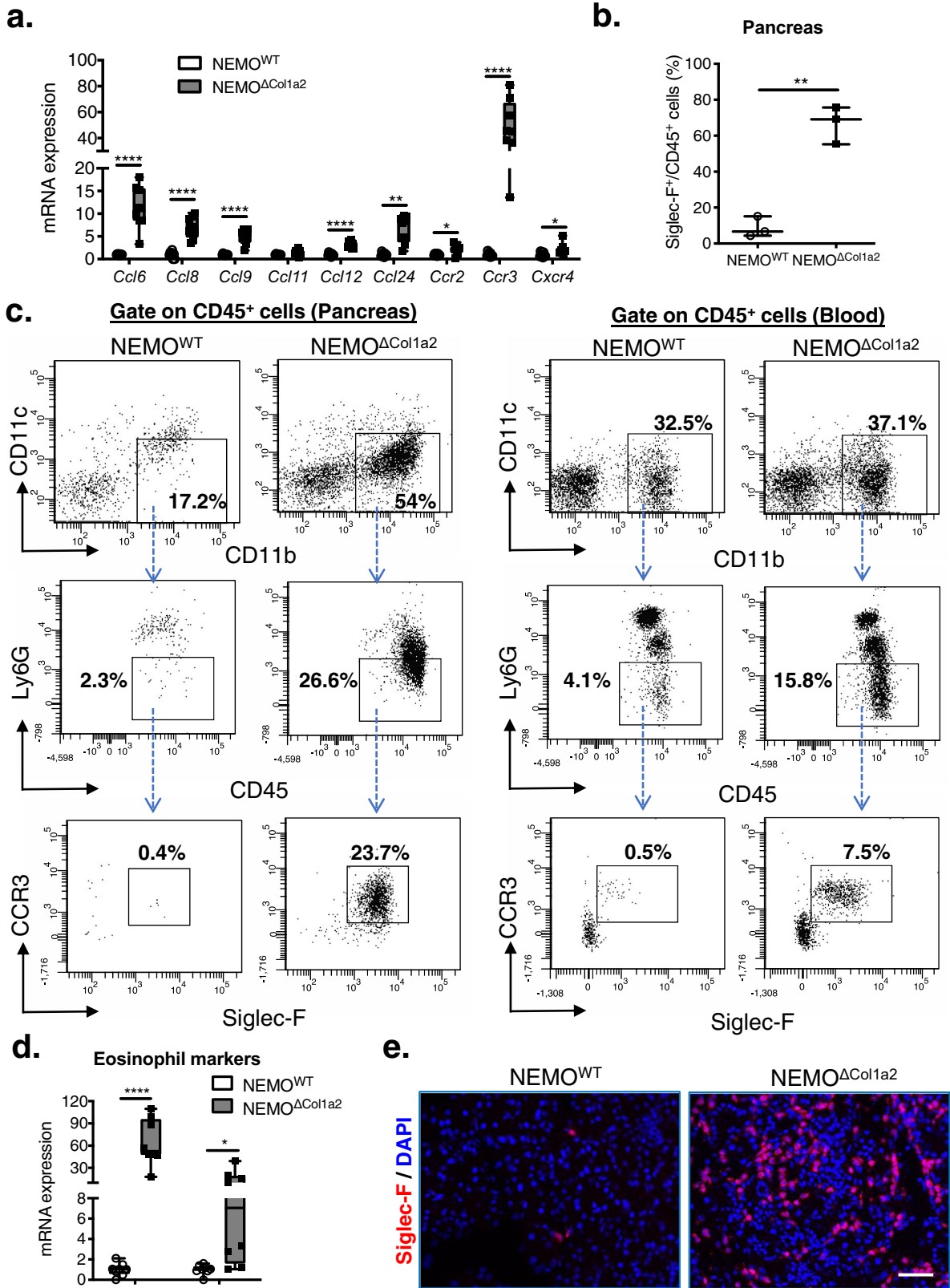

**Fig. 3 Deletion of NEMO in PSCs promotes tissue and peripheral eosinophilia. a** Expression of chemokines and chemokine receptors in pancreata analyzed by qPCR ($n = 8$). **b** Percentage of Siglec-F$^+$/CD45$^+$ cells analyzed by FACS ($n = 3$). **c** FACS analysis illustrating the eosinophil population (gate on CD45$^+$CD11b$^+$CD11c$^−$Ly6G$^−$CCR3$^+$SiglecF$^+$ cells) in pancreas and circulation. **d** Expression of eosinophil markers (*Siglecf* and *Prg2*) detected in the pancreata by qPCR ($n = 8$). **e** Staining of Siglec-F on pancreatic tissues. DAPI: nuclei. Scale bar: 50 µm. Whiskers: Min to Max. *T*-test (two-tailed): *$p < 0.05$; **$p < 0.01$; ****$p < 0.0001$. All n numbers represent biological replicates.

eosinophils in NEMO$^{\Delta Col1a2}$ pancreata throughout the whole organ (Fig. 3e). Interestingly, eosinophilia was not only present in the pancreas, but also in the peripheral circulation (7.5% vs. 0.5%) (Fig. 3c and Supplementary Fig. 4b). Of note, tissue and/or peripheral eosinophilia have been reported in both type I and type II AIP patients and are believed to associate with the pathogenesis of AIP as well as other IgG4-RDs[22,23]. We examined in addition the pancreas samples from the long-term tamoxifen diet-treated groups to confirm if eosinophils were also readily recruited through the same mechanism. Staining of these tissue samples revealed an infiltration of eosinophils together with upregulation of eosinophil markers and the CCL24/CCR3 axis (Supplementary Fig. 5a–c). Therefore, the mechanism leading to eosinophil recruitment to the pancreas was shared between NEMO$^{\Delta Col1a2}$ mice and was independent of caerulein treatment. Caerulein administration, on the other hand, strongly promoted the progression of the disease and increased the severity of pancreatitis.

**Th2-predominant response promotes tissue/peripheral eosinophilia and humoral response.** In human AIP, B and T cells contribute to the initiation and/or progression of the disease. B cell activation with hypergammaglobulinemia is one of the hallmarks of AIP. The Th1/Th2 immune shift may reveal a dynamic change between early and advanced stage of the autoimmune disease, with the Th2-predominant response strongly associated with tissue and/or serum eosinophilia[4]. To study the potential dynamic changes of Th1/Th2 responses, we analyzed the expression of their associated cytokines. An overall increase in the amount of T cells in the NEMO$^{\Delta Col1a2}$ pancreata was observed with CD3 staining (Fig. 4a). Such increase was mainly due to an influx of CD4$^+$ T cells, but not CD8$^+$ T cells. Gene expression analysis showed a significant upregulation of the Th2 cytokines (*Il4* and *Il5*) in NEMO$^{\Delta Col1a2}$ animals, but no significant changes with the Th1 cytokines (*Ifng* and *Il12a*) (Fig. 4b). The Th2-predominant response favored humoral response and also promoted tissue/peripheral eosinophilia in NEMO$^{\Delta Col1a2}$ mice, as IL5 is a known potent mediator for eosinophil activation and maturation[24]. Furthermore, the Th17 subset of T helper cells, as well as the cytokine IL-17, were also found to be increased in the pancreata of NEMO$^{\Delta Col1a2}$ mice (Fig. 4c). Although the role of Th17 cells in other autoimmune diseases like rheumatoid arthritis or psoriasis have been established, their function in AIP still requires further investigations[25]. A study on a small cohort of AIP patients indicates that Th17 cytokines like IL-17 are upregulated in AIP samples and therefore suggesting an association to the pathogenesis of AIP disease[26].

To study the underlying humoral immune response, we first performed B220 staining and indeed revealed an increase in B cell infiltration in the pancreata of NEMO$^{\Delta Col1a2}$ mice (24.2 vs. 0.5 cells/field) (Fig. 4d, e). CD138 staining also confirmed the presence of plasma cells in the NEMO$^{\Delta Col1a2}$ pancreata. Similar to the animals treated with only tamoxifen diet, circulating IgM and IgG levels were again significantly elevated in NEMO$^{\Delta Col1a2}$ mice at different time points (Fig. 4f). This is in line with the observation of increased plasma cells in the pancreata. Further analysis of serological markers of autoimmune disease again indicated that ANA levels were significantly higher (Fig. 4g). Taken together, NEMO$^{\Delta Col1a2}$ mice displayed a Th2-predominant response and an active humoral response which help to sustain or promote the autoimmune phenotype.

**Ablation of NEMO in PSCs leads to stronger fibrosis after caerulein treatment.** We next analyzed the level of fibrosis in the pancreata after repetitive episodes of caerulein-induced pancreatitis. AZAN trichrome staining confirmed a stronger fibrosis

in the pancreata of NEMO$^{\Delta Col1a2}$ mice after caerulein treatment (17.7% vs. 2%), while no prominent connective tissue deposits was observed in the saline-treated groups (Fig. 5a, b). Using the reporter mouse lines (NEMO$^{WT.RFP}$ and NEMO$^{\Delta Col1a2.RFP}$), we found that RFP-expressing cells were seen only in cells at the interstitial area of the pancreas but absent from the epithelial and endothelial compartments (Supplementary Fig. 6). RFP expression did not co-localize with the CD45$^+$ infiltrating leukocytes, but with the PSCs which expressed αSMA, suggesting an activated myofibroblast-like phenotype (Fig. 5c). We next dissected the related mechanisms leading to the pronounced pancreatic fibrosis in NEMO$^{\Delta Col1a2}$ mice. Caerulein treatment led to increased expression of extracellular matrix (ECM) components e.g. *Col1a1*, *Col1a2*, *Col3a1* and *Fn1* (Fig. 5d). NEMO$^{\Delta Col1a2}$ mice showed further upregulation of these genes, including cytokines known to promote fibrogenesis i.e. TGFβ, therefore resulting stronger fibrosis in their pancreata. In addition, we found that the pro-inflammatory cytokine interleukin-6 (IL-6) was 10-fold upregulated in the NEMO$^{\Delta Col1a2}$ pancreata (Fig. 5e). An important role of IL-6 has been established in different inflammatory disorders including autoimmune diseases[27,28]. IL-6 exerts its activity through binding to the IL-6 receptors to activate the downstream pathways including the signal transducer and activator of transcription 3 (STAT3) signaling and mitogen-activated protein kinases (MAPKs) pathways[29]. We therefore also analyzed the activation status of STAT3 and MAPKs in NEMO$^{WT}$ and NEMO$^{\Delta Col1a2}$ mice after experimental pancreatitis. Interestingly, the level of p-STAT3 was found significantly upregulated in the NEMO$^{\Delta Col1a2}$ pancreata, indicating an increased activity of the IL-6/STAT3 signaling (Fig. 5f, g). For MAPKs, the basal levels (saline-treated groups) of p38 and p-p38 were already elevated in NEMO$^{\Delta Col1a2}$ animals compared to the NEMO$^{WT}$ animals (Fig. 5h). This could be caused by a progressive change as a result of NEMO deletion in the PSCs. In the caerulein-treated groups, though p-p38 and p-ERK levels displayed high variabilities, the p-JNK level showed a consistent upregulation in NEMO$^{\Delta Col1a2}$ pancreata. These results suggested a contribution of IL-6 in the AIP phenotype in NEMO$^{\Delta Col1a2}$ mice through the activation of STAT3 and MAPK pathways.

**Enrichment of autoimmune gene signatures in NEMO$^{\Delta Col1a2}$ pancreata resembling the AIP patients.** To further corroborate the observations in the NEMO$^{\Delta Col1a2}$ animals with patient data, we performed a cross-species comparison between NEMO$^{\Delta Col1a2}$ mice and AIP patients. Firstly, we performed microarrays on pancreas samples of NEMO$^{WT}$ and NEMO$^{\Delta Col1a2}$ animals. Similar to what we observed in qPCR analyses, NEMO$^{\Delta Col1a2}$ animals displayed an upregulation of various genes involved in leukocyte migration (Fig. 6a). Using GSEA, we were able to identify a list of enriched gene signatures in NEMO$^{\Delta Col1a2}$ animals in comparison to NEMO$^{WT}$ animals. Interestingly, gene signatures associated with autoimmune diseases (e.g. Autoimmune Thyroid Disease and Lupus Erythematosus) and other inflammatory disorders (e.g. Allograft Rejection and Asthma) were found enriched in NEMO$^{\Delta Col1a2}$ animals (Fig. 6b and Supplementary Fig. 7a). To corroborate these findings with human data, we applied the same approach to analyze microarray data from AIP patients[30]. By comparing AIP and alcoholic chronic pancreatitis (AlCP) patients, we obtained a list of characteristic signatures for AIP. Interestingly, the cross-species comparison revealed that very similar genes were enriched in the human AIP and the murine NEMO$^{\Delta Col1a2}$ samples. The signatures observed in NEMO$^{\Delta Col1a2}$ animals were found enriched in a similar manner in AIP patients (Fig. 6c and Supplementary Fig. 7b). By assessing the whole list of KEGG and Hallmark gene sets from the Molecular Signature Database, we found a high degree of overlap

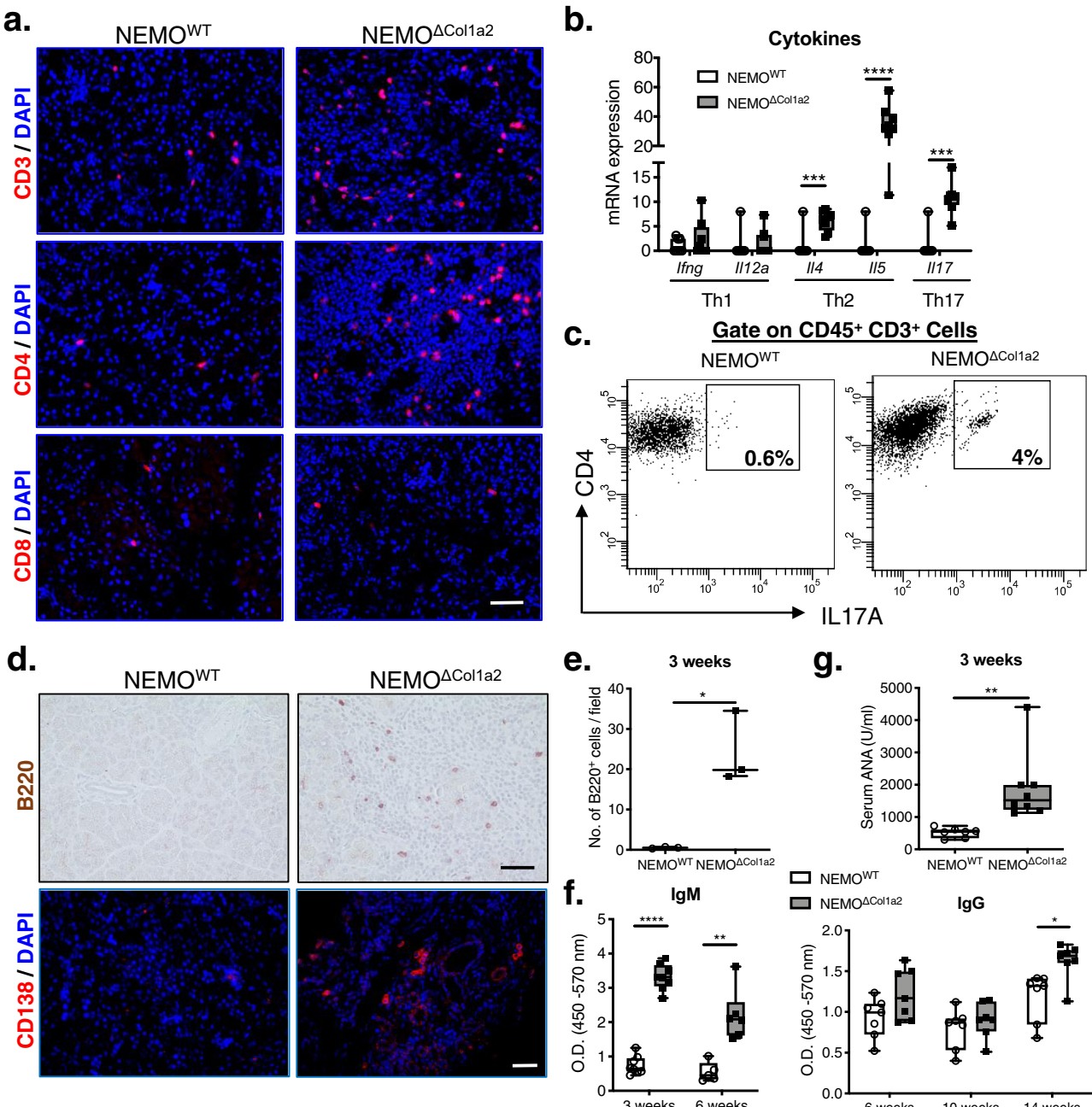

**Fig. 4 Th2-predominant response promotes tissue/peripheral eosinophilia and humoral response. a** CD3, CD4 and CD8 stainings on pancreatic tissues. Scale bar: 50 µm. **b** Expression of Th1, Th2 and Th17 cytokines analyzed by qPCR ($n = 8$). **c** FACS analysis showing an increase in the CD45+CD3+CD4+IL17A+ population in the NEMO$^{\Delta Col1a2}$ pancreata. **d** B220 and CD138 stainings showing increased B cells/plasma cells in the NEMO$^{\Delta Col1a2}$ pancreata. Scale bar: 50 µm. **e** Quantification of B220+ cells per field ($n = 3$). **f** Measurement of the circulating IgM (3 weeks: $n = 8$; 6 weeks: $n = 6$) and IgG ($n = 7$ for all time points) levels by ELISA on serum and the absorbance was determined. **g** Serum ANA levels measured by ELISA ($n = 8$). Whiskers: Min to Max. T-test (two-tailed): *$p < 0.05$; **$p < 0.01$; ***$p < 0.001$; ****$p < 0.0001$. All n numbers represent biological replicates.

between NEMO$^{\Delta Col1a2}$ and AIP patients. In fact, over 90% of the enriched gene sets in AIP patients were also enriched in NEMO$^{\Delta Col1a2}$ animals, though the latter also displayed additional distinctive enriched gene sets (Fig. 6d). Collectively, NEMO$^{\Delta Col1a2}$ animals exhibited several features of autoimmune pancreatitis and with high molecular similarity to human AIP. All these results suggest that the phenotype is very likely autoimmunity in nature.

**Deletion of NEMO in PSCs impairs pancreas recovery after pancreatitis.** We next analyzed the long-term outcome and studied

the animals 10 and 18 weeks after caerulein-induced pancreatitis (Fig. 7a). NEMO$^{WT}$ mice recovered completely from pancreatitis after 18 weeks (Fig. 7b). The NEMO$^{\Delta Col1a2}$ group, however, showed a sustained fibrosis and prolonged upregulation of pro-fibrogenic genes (Col1a1, Col3a1, Fn1 and Vim) 10 weeks after the injections (Fig. 7c). Some of these genes remained elevated even after 18 weeks (Fn1 and Vim). Though, from histology, eosinophil infiltration was reduced comparing to the 3-week time point (Fig. 7d), the expression of eosinophil markers (Siglecf and Prg2) and chemokines (Ccl6 and Ccl24) was still significantly higher comparing to the NEMO$^{WT}$ mice at these later time points. Structures of metaplastic

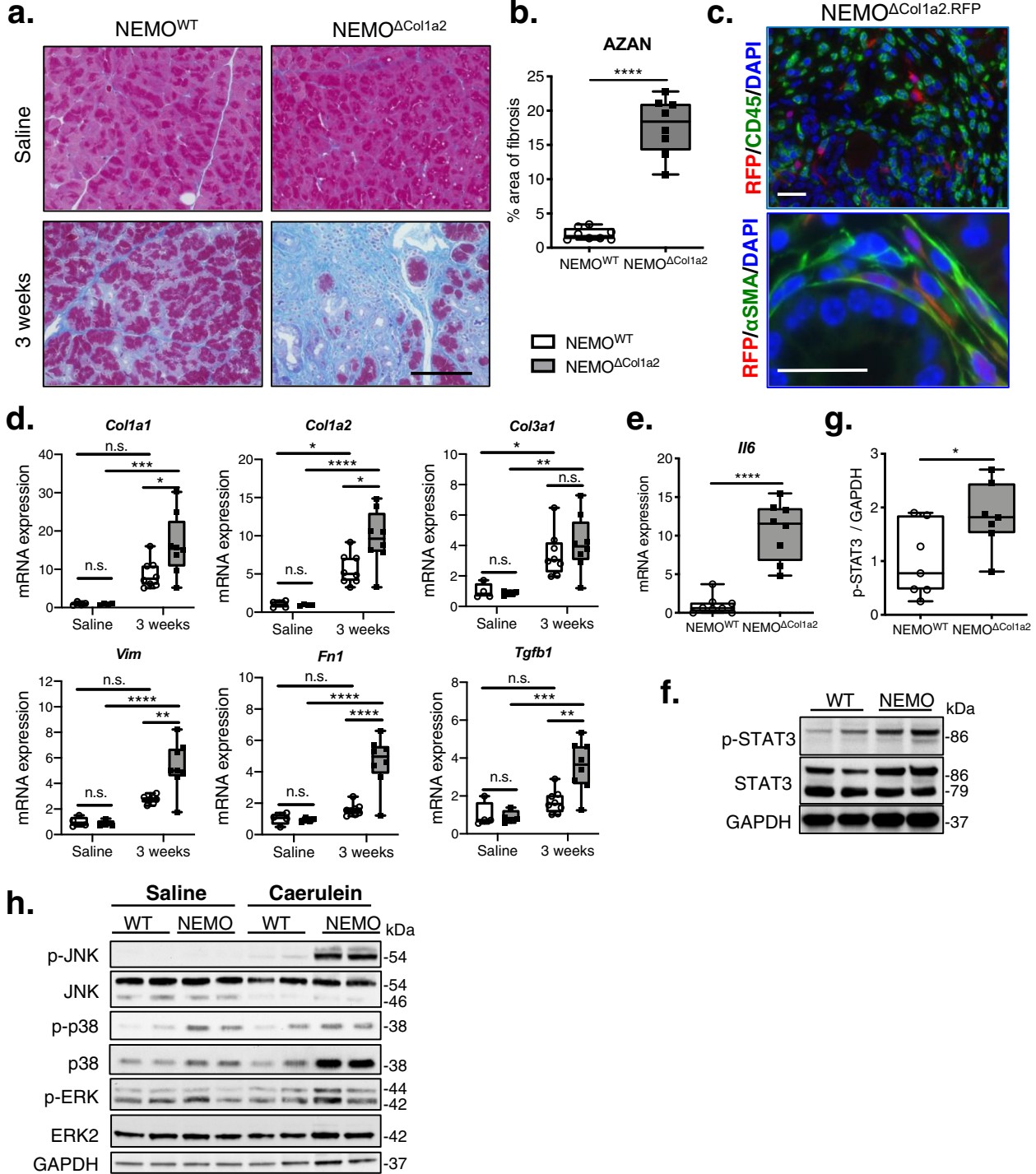

**Fig. 5 Ablation of NEMO in PSCs leads to stronger fibrosis after caerulein-induced pancreatitis. a** AZAN trichrome staining showing an increase in extracellular matrix deposits (blue). Scale bar: 100 μm. **b** Quantification of AZAN staining as a percentage of area (*n* = 8). **c** Co-staining of RFP/CD45 and RFP/αSMA on the pancreatic tissue from the NEMO$^{ΔCol1a2.RFP}$ reporter mice. Scale bar: 25 μm. **d** Expression of pro-fibrogenic genes analyzed by qPCR (Saline: *n* = 4; 3 weeks: *n* = 8). **e** Expression of *Il6* in pancreata analyzed by qPCR (*n* = 8). **f** Immunoblot of p-STAT3 and total STAT3 levels from pancreatic protein extracts. **g** Ratio of p-STAT3 to GAPDH band intensities (*n* = 7). **h** Immunoblot of components of the MAPK pathway on pancreatic protein extracts. WT: NEMO$^{WT}$; NEMO: NEMO$^{ΔCol1a2}$. Whiskers: Min to Max. (**b, e, g**) *T*-test (two-tailed). **d** One-way ANOVA. *$p < 0.05$; **$p < 0.01$; ***$p < 0.001$; ****$p < 0.0001$. All n numbers represent biological replicates.

ductal lesions were frequently retained in these pancreata even after 18 weeks (71% of NEMO$^{ΔCol1a2}$ mice), though no sign of malignant transformation was found. The pancreas weight was significantly lower at this time point comparing to the fully regenerated NEMO$^{WT}$ pancreata (Fig. 7e). Our results showed that animals with NEMO deletion in PSCs failed to suppress the excessive production of ECM components and pro-inflammatory mediators over a long time span. The fact that NEMO$^{ΔCol1a2}$ mice did not recover from caerulein-induced pancreatitis suggested that the disease was not self-limiting.

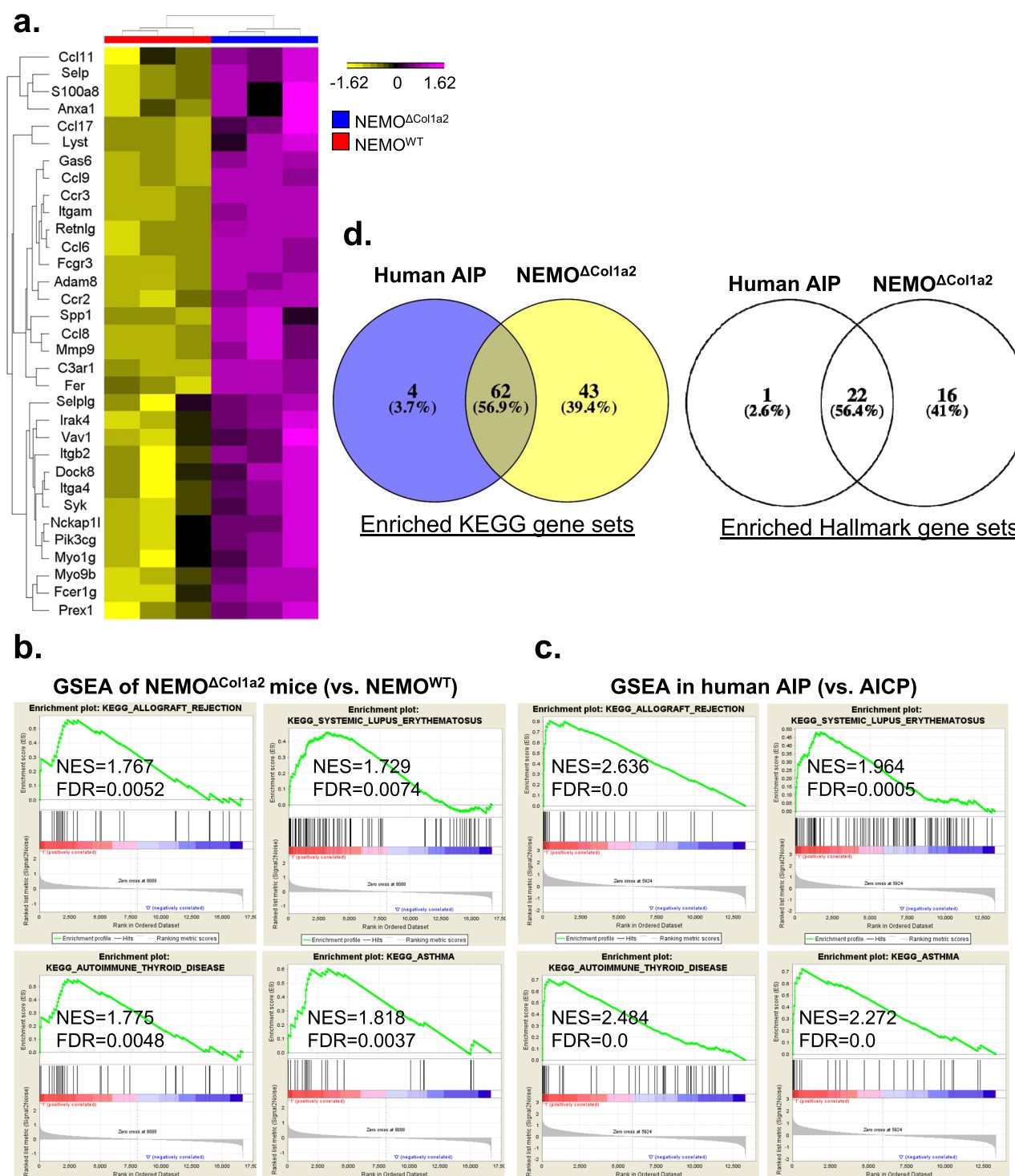

**Fig. 6 Enrichment of autoimmune gene signatures in NEMO$^{\Delta Col1a2}$ pancreata resembling the AIP patients.** DNA microarray performed on pancreatic samples from NEMO$^{WT}$ and NEMO$^{\Delta Col1a2}$ mice from the 3-week time point. **a** Heatmap illustrating genes from the "GO Leukocyte Migration" gene set. **b** GSEA performed on the microarray results from the pancreatic samples of NEMO$^{\Delta Col1a2}$ and NEMO$^{WT}$ mice ($n = 3$). **c** GSEA analysis of microarray data from AIP patients ($n = 4$) using AICP patients ($n = 5$) as a reference. **d** Venn diagrams showing the common GSEA enriched gene sets from the KEGG and Hallmark gene set database (FDR $\leq 0.25$). NES Net enrichment score, FDR False discovery rate. All n numbers represent biological replicates.

**PSC$^{\Delta NEMO}$ cells promote eosinophilia through the secretion of CCL24.** To analyze the molecular changes within PSCs after NEMO deletion (PSC$^{\Delta NEMO}$), we isolated PSCs at the 3-week time point after caerulein and tamoxifen injections. To access the purity of the cell fraction, we stained the cells and found that all isolated cells showed vimentin expression but absence of

CD45 (Fig. 8a). All cells had a fibroblastoid morphology and expressed nestin, another marker of PSCs, in culture (Fig. 8b, c). Quantification of RFP expression indicated that 56% of these cells were nestin$^+$/RFP$^+$. Deletion of NEMO was confirmed at mRNA and protein levels, with PSC$^{\Delta NEMO}$ cells showing 76% and 92% reduction respectively (Fig. 8d, e). Thus, it appears

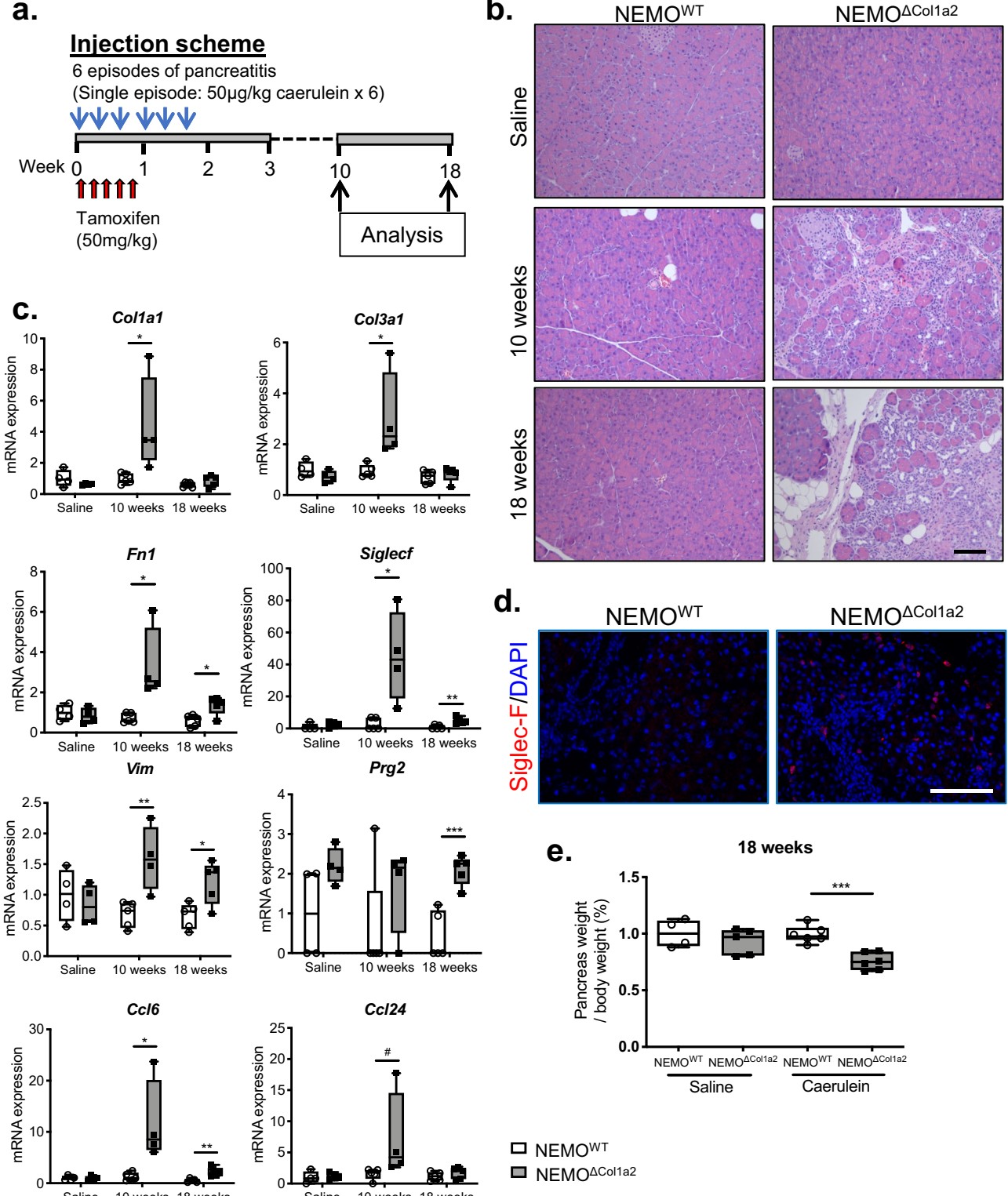

**Fig. 7 NEMO$^{\Delta Col1a2}$ mice demonstrate an impaired recovery after pancreatitis. a** NEMO$^{\Delta Col1a2}$ and NEMO$^{WT}$ mice injected with tamoxifen and caerulein were studied over 18 weeks. **b** Pancreatic histology showing a complete recovery in the NEMO$^{WT}$ group, while the NEMO$^{\Delta Col1a2}$ pancreata still displayed inflammation, fibrosis and structures of metaplastic ductal lesions. Scale bar: 100 μm. **c** Expression of pro-fibrogenic genes and eosinophil markers analyzed by qPCR ($n \geq 4$). **d** Siglec-F staining showing the presence of eosinophils in the pancreata of NEMO$^{\Delta Col1a2}$ mice (18 weeks). Scale bar: 100 μm. **e** Ratio of pancreatic weight to the body weight ($n \geq 4$). Whiskers: Min to Max. T-test (two-tailed): *$p < 0.05$; **$p < 0.01$; ***$p < 0.001$. Mann–Whitney (two-tailed): #$p < 0.05$. All n numbers represent biological replicates.

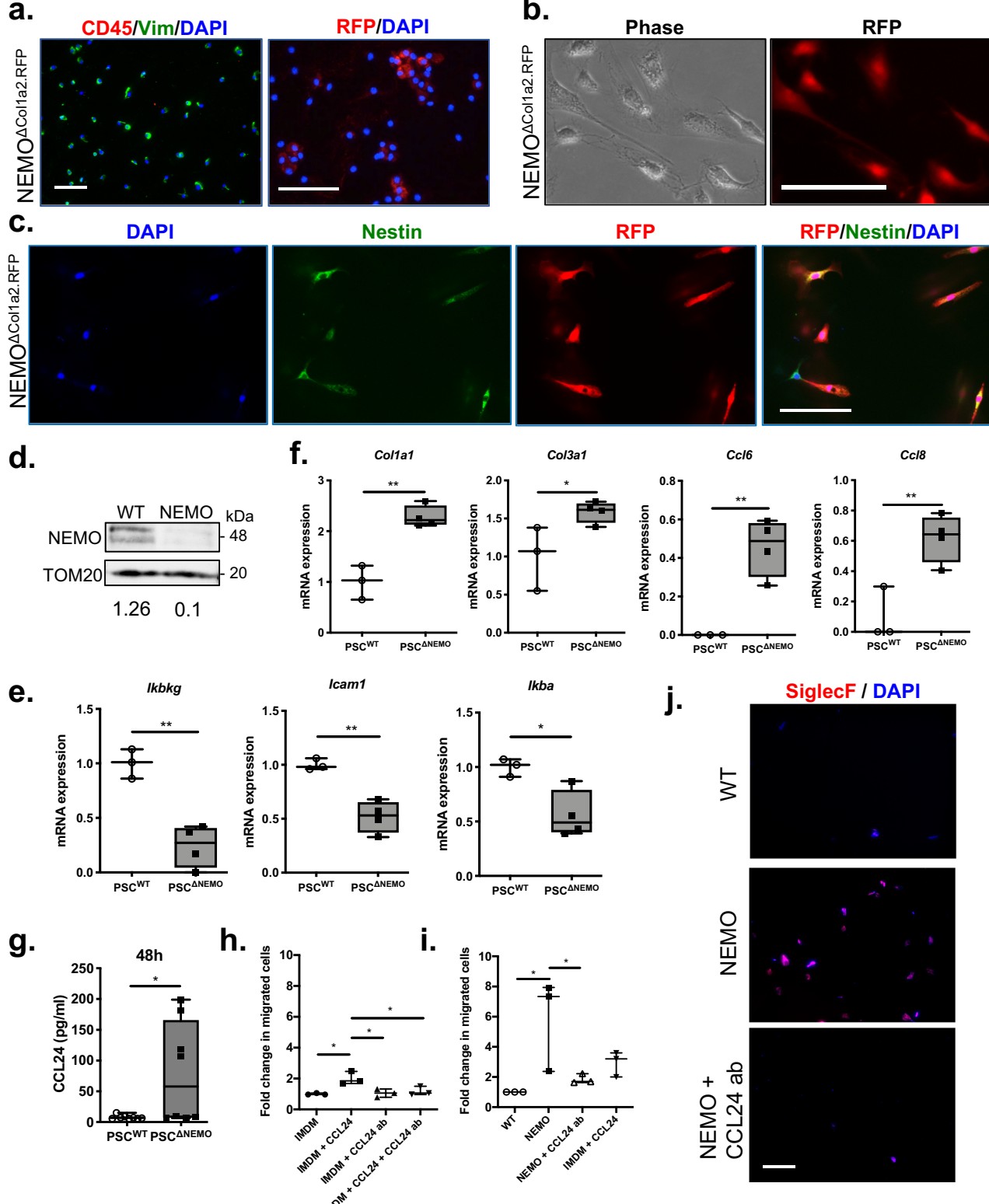

**Fig. 8 PSC$^{\Delta NEMO}$ cells promote eosinophilia through the secretion of CCL24. a** Stainings of Vim, CD45 and RFP on freshly purified PSCs isolated from NEMO$^{\Delta Col1a2.RFP}$ mice (3-week time point). **b, c** PSCs formed fibroblastoid morphology and express RFP. RFP and nestin stainings on PSCs after 24 h in culture. **d** Immunoblot of NEMO on PSC protein extracts. TOM20: loading control. WT: PSC$^{WT}$; NEMO: PSC$^{\Delta NEMO}$. **e, f** Gene expressions in PSCs analyzed by qPCR (PSC$^{WT}$: $n = 3$; PSC$^{\Delta NEMO}$: $n = 4$). **g** Measurement of CCL24 level by ELISA from culture media collected at 48 h ($n = 8$). Whiskers: Min to Max. $T$-test (two-tailed): *$p < 0.05$; **$p < 0.01$. **h, i** Transwell migration assay indicating the fold change of the immune cells towards different media. Media: PSC$^{WT}$ (WT) and PSC$^{\Delta NEMO}$ (NEMO) conditioned media (72 h); media with an addition of CCL24 ligand or neutralization with CCL24 antibody (ab). $n = 3$. One-way ANOVA: *$p < 0.05$. **j** Staining of Siglec-F on the transmigrated cells. DAPI: nuclei. Scale bar: 100 µm. All n numbers represent biological replicates.

that the estimation of recombination efficiency based on RFP expression tended to slightly underestimate the knockout efficiency, possibly due to the regulation of RFP expression through the ROSA.26 promoter and the RFP half-life. Consistent with the strong NEMO reduction, isolated PSC$^{\Delta NEMO}$ cells showed a significant downregulation of NF-κB target genes *Icam1* and *Ikba*, indicating an inhibitory effect on the NF-κB signaling (Fig. 8e). Interestingly, PSCs lacking NEMO showed an increase in the expression of several pro-fibrogenic and extracellular matrix (ECM) regulating genes e.g. *Col1a1, Col3a1, Fn1, Timp1* and *Mmp9* when comparing to their NEMO$^{WT}$ counterpart (PSC$^{WT}$) and therefore suggesting a higher activity in tissue fibrogenesis and ECM modifications (Fig. 8f and Supplementary Fig. 8). While *Ccl6* and *Ccl8* expressions were almost absent in PSC$^{WT}$ cells, strong upregulation was observed in the case of NEMO deletion. As CCL24 is one of the major chemokines which promotes eosinophilia[20], we examined the secretion of this chemokine in cultured PSCs. Purified PSC$^{WT}$ and PSC$^{\Delta NEMO}$ cells were seeded in culture dishes and the medium was collected 24 and 48 h afterwards. Using ELISA, we observed a significant increase in CCL24 secretion by PSC$^{\Delta NEMO}$ cells after 48 h in culture (Fig. 8g). To further test the contribution of PSC$^{\Delta NEMO}$ cells in recruiting eosinophils, we performed a transwell migration assay. Using a recombinant CCL24 (Peprotech 250-22), we first confirmed that the addition of CCL24 to the IMDM medium could promote a transwell migration (Fig. 8h). The effect could be inhibited by the addition of a CCL24 antibody. We next tested if the conditioned media (obtained after 48 h in culture) from PSC$^{WT}$ and PSC$^{\Delta NEMO}$ cells were capable of promoting immune cell migration. Strikingly, the conditioned media from PSC$^{\Delta NEMO}$ cells had much stronger chemotactic effect than PSC$^{WT}$ cells. This chemotactic effect was abolished after neutralization with the CCL24 antibody (Fig. 8i). Finally, we could confirm that the transmigrated cells in response to the conditioned media from PSC$^{\Delta NEMO}$ cells were mostly eosinophils (Fig. 8j). Therefore, PSC$^{\Delta NEMO}$ cells directly contribute to inflammation and fibrosis by increasing the expression of pro-fibrogenic genes and chemokines. Particularly tissue and peripheral eosinophilia appeared to be a direct consequence of an increased secretion of CCL24 by PSC$^{\Delta NEMO}$ cells.

**Corticosteroid therapy attenuates the pancreatitis in NEMO$^{\Delta Col1a2}$ animals.** The high similarity between NEMO$^{\Delta Col1a2}$ animals and AIP patients prompted us to hypothesize that standard treatment regimens for patients could also work on NEMO$^{\Delta Col1a2}$ animals. As corticosteroid treatment is a standard therapy for AIP patients[31], we tested the effect of steroid therapy on NEMO$^{\Delta Col1a2}$ animals. NEMO$^{\Delta Col1a2}$ animals injected with caerulein and tamoxifen received prednisolone through drinking water starting after the first episode of caerulein treatment and over the 3-week experiment period (Fig. 9a). Prednisolone treatment strongly reduced immune cell infiltration and ameliorated the pancreatitis in NEMO$^{\Delta Col1a2}$ animals (Fig. 9b). Metaplastic ductal lesions were completely absent after prednisolone treatment, suggesting that these structures were a result of the extensive inflammation. While NEMO$^{\Delta Col1a2}$ animals without treatment showed a massive infiltration of eosinophils to their pancreata, prednisolone therapy strikingly reduced eosinophil infiltration. This observation was confirmed with a drastic reduction of the eosinophil markers (*Siglecf, Prg2* and *Ccr3*) (Fig. 9c). The expression of various chemokines like CCL6, CCL8 and CCL24 were also significantly downregulated (Fig. 9d). These results indicated a positive response to steroid therapy in NEMO$^{\Delta Col1a2}$ animals, as in AIP patients.

## Discussion

Until today, the immunologic triggers and the pathogenesis of AIP remain mostly unclear. Experimental AIP models have suggested the involvement of autoreactive T cells or the aberrant expression of lymphotoxin α and β in acinar cells as triggers[4,19,32,33]. The current study demonstrates that defects in the NF-κB signaling in PSCs may also be a cause for AIP. A full-blown disease developed rapidly in NEMO$^{\Delta Col1a2}$ animals within two weeks of caerulein administration. This coincides with the activation kinetics of PSCs since parenchymal necrosis and inflammation are known to be the pre-requisites of their activation[34]. The spontaneous inflammation in their pancreata as a result of long-term NEMO deletion further supported their delayed activation without parenchymal damages. Other cells like macrophages may also play an important role in the pathogenesis of the disease, due to their function in the early stages of pancreatic inflammation which precedes PSC activation. We and others have shown before that macrophages belong to the early infiltrating cell population during experimental pancreatitis[7,35]. Macrophage infiltration is prominent in NEMO$^{\Delta Col1a2}$ mice receiving long-term tamoxifen food treatment. Furthermore, infiltrating myeloid cells have been reported to be one of the source of TGF-β secretion during experimental chronic pancreatitis and therefore contributing to PSC activation[8]. Other parameters like premature activation of trypsinogen and activation of NF-κB in acinar cells are also early events of pancreatitis[36]. Studies focusing on the impact of these parameters in AIP will help to understand the complex interactions among the different cell populations in AIP initiation and progression.

Common features shared among other experimental AIP models, as well as in NEMO$^{\Delta Col1a2}$ animals, include B and T cell infiltrations to the pancreas, vasculitis, presence of autoantibodies and elevated circulating immunoglobulins[19,32,37]. Though ANA levels are clearly elevated in NEMO$^{\Delta Col1a2}$ animals, the linkage between the disruption of the NF-κB signaling in PSCs and the generation of autoantibodies is unclear. Whether the presence of autoantibodies is a primary or secondary immunologic stimulus still requires further elucidation. The autoimmune phenotype caused by a lack of NF-κB signaling in PSCs in this study, together with other studies by suppressing the regulatory T-cells or by overexpressing lymphotoxin α and β, may suggest a similar disease with different immune triggers. One limitation of using rodent models to study human AIP is a lack of the IgG4 subtype of immunoglobulin. As AIP is defined as one of the IgG4-RDs, massive infiltration of IgG4$^{+}$ plasma cells is a hallmark for histological diagnosis[4]. Often, this is associated with an elevated serum IgG4 level. However, the role of high IgG4 level in human IgG4-RD has not been established, as IgG4 subtype is a poor trigger of the complement system due to their low affinities to C1q[38]. IgG4 antibodies also function as "blocking antibodies" which promote anti-inflammatory effects and tolerance to allergens[39,40]. Murine IgG1 exhibits a similar property as human IgG4 in inhibiting the classical complement pathway[41]. Therefore, the changes in these immunoglobulins are likely a negative feedback mechanism in response to the inflammatory conditions rather than a causal effect to the disease.

Although constitutive activation of IKK/NF-κB signaling in acinar cells is known to induce pancreatitis, key protective functions of this pathway in reducing cell death and promoting regeneration after injury have also been established[7–11]. In β cells, IKK/NF-κB is required to prevent apoptosis in an autoimmune diabetic mouse model[14]. In the context of pancreatic cancer, PSCs lacking functional NF-κB (p50-/-) promotes infiltration of activated cytotoxic T lymphocytes and suppresses tumor growth[42]. The current study focuses on pancreatic stellate cells and highlights the immunomodulatory role of NF-κB signaling in these

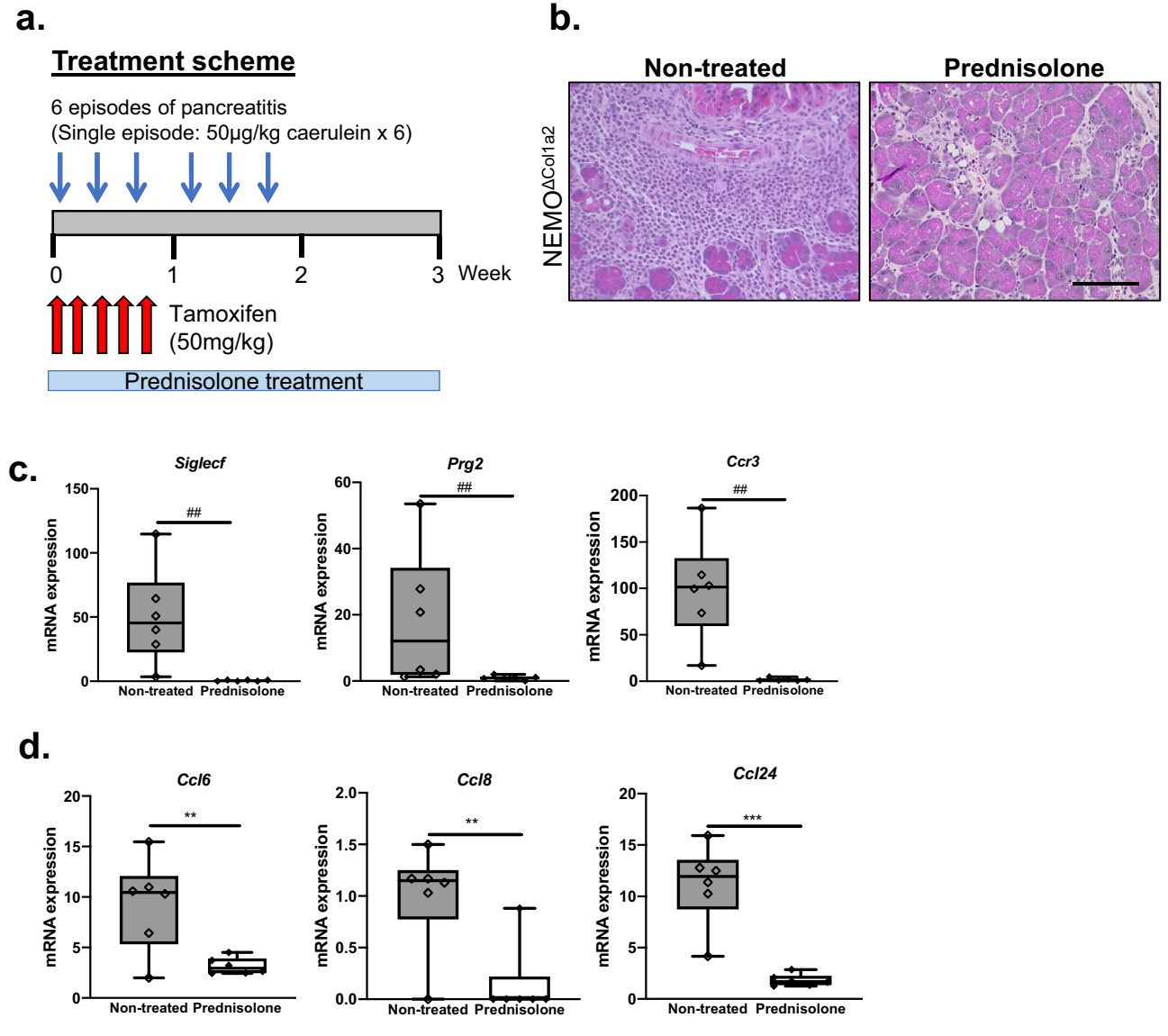

**Fig. 9 Treatment of prednisolone attenuates the AIP phenotype in NEMO$^{\Delta Col1a2}$ mice. a** Scheme of prednisolone treatment. **b** Pancreatic histology of NEMO$^{\Delta Col1a2}$ mice with or without prednisolone treatment. Scale bar: 100 μm. **c, d** Expression of eosinophil markers and chemokines assessed by qPCR (*n* = 6). Whiskers: Min to Max. *T*-test (two-tailed): \*\**p* < 0.01; \*\*\**p* < 0.001. Mann–Whitney (two-tailed): ##*p* < 0.01. The n number represents biological replicates.

cells during pancreatitis. As we have not seen extensive apoptosis in PSCs, the protective function of IKK/NF-κB in PSCs appears to be highly reliant on its role in limiting the production of several inflammatory mediators and reducing infiltrating cells. Although NF-κB is known to be a master regulator of inflammatory genes including various chemokines like MCP-1, KC and RANTES[43], its inhibition in PSCs also lead to upregulation of other chemokines like CCL6, CCL8 and CCL24 which are not target genes of NF-κB. Our results may suggest a crosstalk between NF-κB and other pathways and therefore reveal an important role of PSCs as active immune modulators during pancreatitis. The canonical NF-κB signaling in these cells is essential to counteract an "overshooting" inflammatory response. From the analyses of PSCs and pancreas tissues from NEMO$^{\Delta Col1a2}$ animals, we consistently observed an increased expression of IL-6 and activation of the downstream pathways. This is consistent with earlier studies that have revealed the significant role of the IL6-STAT3 signaling in the development of inflammatory disorders, including pancreatitis, and its inhibition proves to be beneficial in

autoimmune diseases like rheumatoid arthritis and multiple sclerosis[44–46].

In summary, this study highlights the role of PSCs as central regulators of inflammatory responses and an intact NF-κB signaling in these cells is essential to prevent the development of spontaneous pancreatitis of an autoimmune nature. PSC$^{\Delta NEMO}$ cells contribute to tissue and peripheral eosinophilia through a massive secretion of CCL24.

## Methods
**Mice.** NEMO$^{\Delta Col1a2}$ mice (C57BL/6) were generated by crossing Col1a2-Cre.ERT mice with those carrying floxed *Ikbkg* allele[6,47,48]. The reporter lines NEMO$^{WT.RFP}$ and NEMO$^{\Delta Col1a2.RFP}$ were created by breeding NEMO$^{WT}$ (Col1a2-Cre.ERT-tg + /NEMO$^{WT}$) and NEMO$^{\Delta Col1a2}$ (Col1a2-Cre.ERT-tg + /NEMO$^{fl}$) animals with ROSA26.LSL-RFP mice respectively[49]. Mice from both genders, with age from 6 to 12 weeks, were used in this study. Since NEMO is X-linked, males were hemizygous (NEMO fl) and females were homozygous (NEMO fl/fl). Both genders developed a similar phenotype. Gender and age matched littermates were used as controls. The sample size for each experiment is shown in the figure legends of each graph. Detail genotypes of mice used in this study: NEMO$^{WT}$: Col1a2-Cre.ERT-tg + /NEMO$^{WT}$ or Col1a2-Cre.ERT-tg-/NEMO$^{fl}$; NEMO$^{\Delta Col1a2}$: Col1a2-

Cre.ERT-tg + /NEMO$^{fl}$; NEMO$^{WT.RFP}$: Col1a2-Cre.ERT-tg + /NEMO$^{WT}$/ROSA26.LSL-RFP; NEMO$^{\Delta Col1a2.RFP}$: Col1a2-Cre.ERT-tg + /NEMO$^{fl}$/ROSA26.LSL-RFP.

**Secretagogue-induced pancreatitis**. Caerulein (#H-3220) was obtained from Bachem (Switzerland). Stock caerulein solution (0.5 mg/ml) was prepared in sterile water and stored at −20 °C until use. Working caerulein solution was prepared freshly on each injection day in 0.9% NaCl solution to make a 10 μg/ml solution. Repetitive episodes of pancreatitis (three episodes per week, i.e. Monday, Wednesday and Friday, over two weeks) were induced in animals with each episode consisting of six hourly injections (50 μg/kg) on each experiment day. Control groups received 0.9% NaCl solution.

**Tamoxifen treatment**. To induce the activity of Cre recombinase, tamoxifen (Santa Cruz #sc-208414) was prepared in 12.5% vol/vol ethanol/oil and injected subcutaneously (50 mg/kg) for five days in the first week of experiment. For tamoxifen food treatment, animals had free access to the tamoxifen diet over a period of 18 weeks. Tamoxifen diet (400 mg/kg) was purchased from Genobios (France).

**Prednisolone treatment**. Predinsolone (Sigma P6004) was prepared in ethanol at a concentration of 30 mg/ml and added to 500 ml of drinking water. Prednisolone solution was renewed every other day. For the treatment group, animals received prednisolone starting on the first day of caerulein injection (after the first episode) until the end of the experiment (throughout the 3-week experiment period). Control animals received normal drinking water with the equivalent amount of ethanol.

**Histology, immunohistochemistry (IHC), immunofluorescence (IF) and microscopy**. For paraffin sections, pancreata were fixed in 4% neutral buffered formalin for at least 6 h and then proceeded to dehydration and embedded with paraffin. Paraffin sections of 3 μm thickness were used in each staining. For cryosectioning, tissue was first snap-frozen in liquid nitrogen and embedded in tissue freezing medium (Tissue Tek). Cryosections of 4 μm were freshly prepared for each staining. Hematoxylin and eosin (H&E) and Alcian blue stainings were performed by the standard protocol. For Heidenhain's AZAN trichrome staining, sections were stained according to the staining kit protocol provided by the manufacturer (Morphisto). For Immunohistochemical (IHC) and immunofluorescence (IF) stainings of paraffin-embedded tissue, heat-mediated antigen retrieval (pressure cooker, 10 min) was performed with citrate buffer (pH 6) after deparaffinization and rehydration. For IHC, sections were treated with 3% $H_2O_2$ for at least 10 min to neutralize the endogenous peroxidase activity. The sections were then blocked with 5% BSA/PBS solution for one hour before adding the primary antibodies. For cryosections, the tissue sections were freshly prepared and fixed with 4% formalin then followed by permeabilization with 0.1% Triton-X/PBS solution. The same procedure applied as in paraffin-sections for blocking and antibody applications. A list of antibodies is provided in Supplementary Table 1. For quantifications of staining, six fields of view/section were analyzed with the BZ Analyzer software or imageJ.

**Isolation of infiltrating cells from pancreas**. Pancreata were carefully excised and chilled in Hank's Balanced Salt Solution (HBSS). Two ml of the collagenase D (Roche #11088866001) solution (4 mg/ml collagenase D, 2.5 mg/ml DNase I, HBSS/HEPES) was injected to each pancreas and then incubated at 37 °C with shaking for 15 min. Afterwards, the pancreas was divided into fine pieces with scissors and incubated at 37 °C for another 15 min. The dissociated tissue solution was passed through a 40 μm nylon mesh and then subjected to density-gradient centrifugation with Histopaque-1077 (Sigma #10771) and Histopague-1119 (Sigma #11191). Cells were collected at the interphases between the different solutions and used in flow cytometry.

**Flow cytometry**. Fluorescence-activated cell sorting (FACS) was performed with BD FACSCantoII. Approximately 1 million cells were resuspended in 100 μl of staining buffer (1% BSA, 1 mM EDTA, 1 μg/ml DNase I, HBSS/HEPES). After blocking with 1 μg of Mouse BD Fc Block$^{TM}$ (BD Pharmingen #553141), the cells were incubated with antibody cocktail solutions for 30 min at 4 °C. A list of FACS antibodies was included in Supplementary Table 2. The cells were washed thoroughly with the staining buffer and analyzed with BD FACSCantoII. The FACS results and graphs were prepared with the BD FACSDiva software.

**Isolation of PSCs**. PSC cell fractions were isolated by density-gradient centrifugation using Nycodenz solution following the protocol developed by Dr. Minoti Apte with modifications[50]. Pancreata were digested with an enzymatic solution (1.3 mg/ml Collagenase P (Roche #11213857001), protease (Sigma #P5147), 0.01 mg/ml DNase I, GBSS with NaCl). The dissociated tissue solution was subjected to density-gradient centrifugation with 11.4% Nycodenz solution (Progen Biotechnik #1002424) prepared in GBSS without NaCl. Cells were collected at the interphase of the solutions. For purification of PSCs, the collected cells

were incubated with CD45 microbeads (Miltenyi Biotec #130-052-301) and passed through MACS LD columns (Miltenyi Biotec #130-042-901). The purified cells were either analyzed by flow cytometry or cultured in IMDM (PAN-Biotech #P04-20050) supplemented with 10% FBS (Gibco #10270) and 1% Penicillin/Streptomycin (Gibco #15140).

**Microarray**. Transcriptomic profiling was performed in the Genomics-Core Facility of the University of Ulm using the Mouse Gene 2.0 ST Array (Affymetrix). Total RNA extracts (1 μg/sample) from the pancreata of NEMO$^{WT}$ and NEMO$^{\Delta Col1a2}$ animals were used in this transcriptome analysis ($n = 3$). The dataset was submitted to ArrayExpress (Accession# E-MTAB-7274). For analyzing the transcriptome profiles in patients, microarray data from AIP and alcoholic chronic pancreatitis (AlCP) patients was obtained from ArrayExpress (E-MEXP-804)[30]. The data was analyzed with Partek Genomic Suite.

**Gene set enrichment analysis (GSEA)**. To perform GSEA analysis on the microarray data, the datasets from NEMO$^{WT}$ and NEMO$^{\Delta Col1a2}$ animals (E-MTAB-7274), as well as the datasets from autoimmune pancreatitis and alcoholic pancreatitis patients (E-MEXP-804), were analyzed with javaGSEA (3.0) Desktop Application.

**Western blot**. For the extraction of protein, the pancreas tissue was snap-frozen and homogenized by powderization. The total protein extract was obtained with 4% sodium dodecyl sulfate (SDS)/100 mM Tris-HCl. The protein concentration was determined by the BCA Protein Assay Kit (Thermo Scientific). For western blots, 10–20 μg of the protein was prepared with Laemmli buffer and heated at 95 °C for 10 min. For separating proteins, 4–12% SDS-polyacrylamide gradient gels (Invitrogen) were used. A semidry blotter (Bio-Rad) was used to transfer proteins onto nitrocellulose membranes. The membranes were blocked with 5% w/v non-fat dry milk prepared with TBS buffer/0.1% Tween-20 for 1 h. Primary and secondary antibodies were then applied in the subsequent steps. A list of antibodies for western blot is provided in Supplementary Table 3. The band signal was detected by x-ray films. Quantification of the bands was performed using the ImageJ software.

**Quantitative real-time PCR (qPCR)**. The total RNA was isolated from pulverized pancreas tissues according to the RNeasy kit protocol provided by the manufacturer (Qiagen). Complementary DNA (cDNA) synthesis was performed using the Transcriptor High Fidelity cDNA Synthesis Kit (Roche) with 1–2 μg of the extracted mRNA from each sample. Gene expression levels were analyzed by qPCR using LightCycler 480 (Roche). Primers specific to the genes of interest were designed according to the recommendation of the Universal Probe Library (Roche). A list of primers is provided in Supplementary Table 4.

**Serum enzyme activities measurement**. To determine the serum amylase activity, blood was collected from the animals at the indicated time points. Amylase activities were measured using the Reflotron Analyzer with the pancreatic amylase strips (Roche #11200658). Serum lipase activity was determined by the Lipase-PS™ kit (Trinity Biotech #805 A) according to the manual provided by the manufacturer.

**Enzyme-linked immunosorbent assay (ELISA)**. For measuring CCL24 levels in the culture medium, purified PSCs were cultured in IMDM for 48 h. The level of CCL24 was determined by the Eotaxin-2(CCL24) mouse ELISA Kit (Invitrogen #EMCCL24). ELISA kits were also used for measuring the total IgM (Affymetrix #88-50470-22), IgG (Affymetrix #88-50400-22) and ANA (Alpha Diagnostic #5210) levels in serum. All measurements were carried out according to the protocols provided by the manufacturer.

**Transwell migration assay**. PSCs were isolated and purified from NEMO$^{WT}$ and NEMO$^{\Delta Col1a2}$ pancreata using the protocol described above. The isolated PSCs were seeded at a density of 40,000 cells per well in a 12-well culture plate using IMDM medium (10%FBS, 1% P/S, 4 mM Glutamine). The conditioned media were collected after 48 h.

On the day of transwell migration assay, immune cells were isolated from the blood of a test mouse. Briefly, whole blood (1 ml) was collected from vena cava and immediately transferred to 1X RBC lysis buffer (1:15). After 5 min incubation at room temperature, the immune cells were centrifuged at 300 g for 5 min. The supernatant was aspirated and the cells were resuspended in 1 ml of 0.5% BSA/PBS. The cells were then kept on ice until use.

Preparing the transwell plate (5 μm pore size, costar 3421) for migration assay: add to each lower well 600μl of the testing medium followed by insertion of the upper wells which carry the membrane. Preincubate the transwell plate at room temperature for 1 h. List of media prepared: IMDM, IMDM with 200 ng/ml recombinant Murine CCL24 (Peprotech 250-22), IMDM with 0.2 μg/ml CCL24 antibody (R&D MAB528-SP), IMDM with 200 ng/ml recombinant Murine CCL24 and CCL24 antibody, PSC$^{WT}$ conditioned media, PSC$^{\Delta NEMO}$ conditioned media and PSC$^{\Delta NEMO}$ conditioned media with 0.2 μg CCL24 antibody. After

preincubation, transfer 100,000 immune cells to the upper well and incubate the transwell plate at 37 °C for 6 h. At the end of the incubation period, carefully remove the upper well and count the transmigrated cells in the lower well. All conditions were repeated 3 times.

**Study approval**. All animal experiments followed the relevant ethical regulations and were performed in accordance to the German animal welfare legislation with approvals from the responsible organization (Regierungspräsidium Tübingen–Referat 35).

**Statistics and reproducibility**. Statistical analyses were performed using GraphPad Prism 8. Student's $T$-test (two-tailed) was used for comparisons of two groups ($*p < 0.05$, $**p < 0.01$, $***p < 0.001$, $****p < 0.0001$). In case outliers were spotted in sample groups, Mann–Whitney test was used ($\#p < 0.05$, $\#\#p < 0.01$, $\#\#\#\#p < 0.0001$). One-way ANOVA was performed on comparisons with more than two groups ($*p < 0.05$, $**p < 0.01$, $***p < 0.001$, $****p < 0.0001$). For One-way ANOVA test, we assumed equal standard deviations and Tukey's multiple comparasons test was used. All sample sizes for each experiment were described in the figure legend of the corresponding graphs. Data are represented in Box-whisker plot with whiskers indicating the maximum and minimum values.

**Reporting summary**. Further information on research design is available in the Nature Research Reporting Summary linked to this article.

## Data availability

The microarray data was deposited in ArrayExpress with accession number E-MTAB-7274. Original Western blot images are provided in Supplementary Fig. 9. All data used in graphs are provided in Supplementary Data 1.

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

## Acknowledgements

We appreciate the excellent technical assistance from Uta Manfras. We also thank Dr. Hans-Jörg Fehling for generous sharing of the RFP reporter mouse.

## Author contributions

L.K.C. conceived the study concept and design, performed experiments, acquired and analyzed data, and prepared the manuscript. M.T. acquired data, performed experiments and involved in the extensive revision work. M.G. performed experiments and histological analyses. L.D.F. performed data analysis. F.L. contributed to histological analyses. A.K. provided support for intellectual content, as well as critical revision on the manuscript. H.J.M. conceived the study concept and design as well as carried out study supervision. T.W. contributed in the study concept and design, carried out study supervision, provided critical revision on the manuscript and supplied the research funding.

## Funding

## Competing interests

The authors declare no competing interests.
