## [Peer Review File · Communications Biology]

Reviewers' comments:

Reviewer #1 (Remarks to the Author):

This manuscript described the inflammation-triggering effect of PSC-specific knockout of NEMO. This modulation resulted in the exacerbation of spontaneous pancreatitis and caerulein-induced pancreatitis in mouse. The conditional knockout of NEMO resulted in the increased infiltration of eosinophils toward pancreatic tissue, driven by CCL 24. Presented data suggest certain contribution of this molecule to the inhibition of pancreatic inflammation. Authors claimed that this mouse could be a novel mouse model of autoimmune pancreatitis. Unfortunately, experimental methods and presented data are not supporting authors' claim adequately. I agree that this study clarified a novel anti-inflammatory function of PSC during pancreatitis and pancreatic fibrosis. However, obtained data need to be reconsidered whether it could be a real pathogenesis in AIP. At this stage, it is difficult to consider this mouse model as an AIP model. Authors might be better to focus on the mechanistic aspect of this model, rather insisting to be a novel disease model. Followings are my comments.

1.

Authors should confirm the characteristics of AIP's histology. Infiltration of lymphocytes and plasma cells, called LPSP, is typical. Even for type II AIP, eosinophilic infiltration is not frequent. As a starting point, these definitions need to be confirmed.

2.

Serum amylase levels should be confirmed in mice with spontaneous pancreatitis caused by NEMO CKO. Macroscopic image of pancreas also should be presented.

3.

In figure S3c, saline-treated mice showed serum amylase elevation. Condition of mouse treatment needs to be checked.

4.

Authors presented figure 2d for the existence of obliterative venulitis in the mouse model. However, it seems different from obliterative phlebitis seen in AIP. Identification of infiltrating cells also should be carried out.

5.

Mechanistic studies of NEMO-deleted PSC are insufficient. Authors should confirm cell proliferation, migration and ECM production in NEMO-deleted PSC. Effects of PSC-conditioned medium on immune cells must be validated, in combination with neutralizing antibodies for specific factors.

6.

Alteration of intracellular signals in NEMO-deleted PSC need to be validated. Whole tissue lysate includes pancreatic acinar cells that should have altered intracellular signals by inflammation.

7.

Authors stated that this mouse model exhibited key features of AIP in line 268-269, but I cannot agree with this statement. This model represents eosinophil-related pancreatitis model. Authors should clarify to which disease entity this mouse model can be attributed.

8.

Additional knockout of CCL24 will be the best way to confirm that this cytokine is essential for NEMO deletion-induced pancreatitis.

9.

There are several typos such as line 69, there is no space before the brackets. Please check.

Reviewer #2 (Remarks to the Author):

This manuscript describes a genetically engineered mouse model for autoimmune pancreatitis by conditionally blocking the activation of NF κ B pathway in mesenchymal cells. In addition, they show that the lack of mesenchymal NF κ B signaling increases the fibrotic response to induction of chronic pancreatitis (CP) by repeated cerulein treatment. As such, this is a very interesting and important contribution to the field of pancreatitis research. My expertise in immunology is rather limited and therefore I will restrict my review on the non-immunologic parts of this manuscript.

The authors should introduce the correct nomenclature (homepage of Jax lab is a good source) of the Cre-driver, which is B6.Cg-Tg(Col1a2-cre/ERT,-ALPP)7Cpd/J and the correct nomenclature of IKK-gamma/Nemo floxed allele: $I\kappa b\kappa g^{tm1Mpa}$. To call the KO mice NEMO Δ Col1a2 is very misleading, as it implies the lack of Col1a2 in my understanding. I would suggest something like Col1a2-CRE^{ERT};Nemo^{fl}, or just call them cdNemo^{KO} for conditional Nemo knockout.

How have the mice been genotyped? Nemo is X-linked, females in experiments had homozygous or heterozygous deletion?

Was there any sex-specific difference in the extent of AIP or CP?

Fig. S1b (and S6) shows that recombination has taken place, but not in which cells (double staining?) and to which extent. This important information comes to some extent in Fig. 5c and 8a-d (statistics?). I think it would aid in understanding if the characterisation of the recombination efficiency would be put together in a paragraph early in the manuscript.

What is TOM20?

Fig. S1a implies analysis 2 weeks after Tam injection, whereas the figure legend reads 3 weeks after Tam injection.

Looks like there are no tables in the downloadable pdf.

Line 214: "Fig. 3g" should read "Fig. 4g".

Are AIP patients treated with prednisolone before AIP detectable symptoms? If not, the mice should be treated after TAM/Cer induction to better reflect the clinical situation.

Point-by-point response to reviewers

Referee expertise:

Referee #1: Pancreatic Stellate Cells, Autoimmune Pancreatitis, Chronic Pancreatitis

Referee #2: Murine models, Pancreatic cancer, Chronic pancreatitis

Reviewers' comments:

Reviewer #1 (Remarks to the Author):

This manuscript described the inflammation-triggering effect of PSC-specific knockout of NEMO. This modulation resulted in the exacerbation of spontaneous pancreatitis and caerulein-induced pancreatitis in mouse. The conditional knockout of NEMO resulted in the increased infiltration of eosinophils toward pancreatic tissue, driven by CCL 24. Presented data suggest certain contribution of this molecule to the inhibition of pancreatic inflammation. Authors claimed that this mouse could be a novel mouse model of autoimmune pancreatitis. Unfortunately, experimental methods and presented data are not supporting authors' claim adequately. I agree that this study clarified a novel anti-inflammatory function of PSC during pancreatitis and pancreatic fibrosis. However, obtained data need to be reconsidered whether it could be a real pathogenesis in AIP. At this stage, it is difficult to consider this mouse model as an AIP model. Authors might be better to focus on the mechanistic aspect of this model, rather insisting to be a novel disease model. Followings are my comments.

1. Authors should confirm the characteristics of AIP's histology. Infiltration of lymphocytes and plasma cells, called LPSP, is typical. Even for type II AIP, eosinophilic infiltration is not frequent. As a starting point, these definitions need to be confirmed.

Answer to reviewer #1 point 1:

We thank reviewer #1 for bringing up this issue and we agree that it is important to clarify the disease entity in our mouse model. Therefore, we would like to discuss the different observations that convinced us that the phenotype was autoimmunity related.

Firstly, we observed elevated levels of circulating IgM and IgG levels. More importantly, we also observed an elevation of autoantibodies (Fig 1e-f & Fig 4g-f). Due to potential overlapping features with eosinophilic pancreatitis, the presence of autoantibodies in AIP was suggested to be used to distinguish the two entities (Manohar *et al.*, 2021). Secondly, by checking the level of lymphocytes and B cells (B220) /plasma cells (CD138), we observed an increase in all these cells in NEMO^{ΔCol1a2} pancreata (Fig 4a, d & e; also provide with lower magnifications in Letter figure 1a). Thirdly, GSEA on the microarray results from NEMO^{ΔCol1a2} mice showed the top enriched signatures were signatures related autoimmune diseases e.g. Autoimmune Thyroid Disease. This was also consistent with the microarray results from AIP patients (Fig 6 b-c). In contrast, analysis of chronic pancreatitis microarray results from mice injected with caerulein showed the opposite i.e. depleted signatures for Autoimmune Thyroid Disease (Letter figure 1b). Fourthly, NEMO^{ΔCol1a2} mice showed a positive response to prednisolone treatment (Fig 9). Histologically, we observed a strong increase in pancreatic fibrosis and sporadic venulitis. Taken together, these results all support the conclusion that the phenotype observed in NEMO^{ΔCol1a2} pancreata was very likely autoimmunity in nature.

With respect to the point that eosinophilic infiltration is not frequent in AIP, there are reports in the literature showing an association of eosinophilia with AIP diseases. In one of the studies, Wang *et al.* showed that the percentage of peripheral eosinophilia (above 0.5 X 10⁹ /ml) in patients was significantly higher in AIP patients comparing to non-AIP chronic pancreatitis patients, which were 42.9% and 13.3% respectively (Wang *et al.*, 2009). In another study, Sah *et al.* reported that 28% in their AIP patient cohort had peripheral eosinophilia at presentation or during follow-up (Sah *et al.*,

2010). Moderate-to-severe eosinophilic infiltration to the pancreas was observed in 54% of their AIP patient specimens. Although these studies may come from individual centers and rely on small patient cohorts, they should still be relevant to the disease. Since AIP is a rare disease, with a prevalence of less than 1 per 100,000, it is obvious that multicenter or even multinational collaboration studies will further help with our understanding of the disease.

Letter figure 1. a) Staining of CD3 and CD138 on pancreas sections from the 3-week time point. Scale bar: 100µm. b) Microarray data of saline- and caerulein-injected C57BL/6J mice was obtained from ArrayExpress (E-GEOD-41418) (Ulmasov *et al.*, 2013). Gene signatures were analysed by GSEA 4.1.0. Results indicated while inflammation and fibrosis signatures were enriched in the caerulein-injected mice, other signatures like autoimmune thyroid disease were depleted.

2. Serum amylase levels should be confirmed in mice with spontaneous pancreatitis caused by NEMO CKO. Macroscopic image of pancreas also should be presented.

Answer to reviewer #1 point 2:

We thank reviewer #1 for this suggestions. Now the macroscopic images are presented in Fig S2a and the serum amylase levels are provided in Fig S2b.

3. In figure S3c, saline-treated mice showed serum amylase elevation. Condition of mouse treatment needs to be checked.

Answer to reviewer #1 point 3:

We thank reviewer #1 for pointing out this issue. We checked the amylase levels of individual mice, particularly the saline-treated NEMO^{WT} mice. To better visualize the individual values, we presented the graph in scatter-plot format (Letter figure 2). We found that one of the mice in the saline-treated NEMO^{WT} group had a relatively high amylase level. However, the level was still much lower than animals at the acute phase after caerulein injections (12 hours) which was often over 10,000 U/L (Chan *et al.*, 2017)(Neuhöfer *et al.*, 2013). Importantly, the pancreas histology of this mouse showed no abnormalities. In addition, most of the caerulein-injected mice also did not show a drastic increase in their amylase levels at the 3-week time point, except in two of the NEMO^{Col1a2} mice. One explanation was that they had passed the acute phase of the effect induced by caerulein. Overall we did not see a significant change at the 3-week time point between NEMO^{WT} and NEMO^{Col1a2} mice.

Letter figure 2. Characterization of animals after caerulein and tamoxifen injections at the 3-week time point. Scatter plot representation of the amylase levels measured from serum (n≥4). Mean ± SD.

4. Authors presented figure 2d for the existence of obliterative venulitis in the mouse model. However, it seems different from obliterative phlebitis seen in AIP. Identification of infiltrating cells also should be carried out.

Answer to reviewer #1 point 4:

We thank reviewer #1 for this comment. We agree that the picture shown in fig 2d had signs of venulitis, but may not be considered as obliterative phlebitis since the lumen of the vein was still visible. Since in our case, even venulitis was a sporadic event, we observed it only in one out of the eight NEMO^{Col1a2} pancreata which completed the 3-week caerulein and tamoxifen protocol and also in one out of the six NEMO^{Col1a2} mice which developed spontaneous pancreatitis. Characterization of the infiltrating cells throughout the pancreas was done and shown in Fig 3-4 either by immunostaining or by flow cytometry. Here we show stainings around one of the potentially affected veins (Letter figure 3). Since it is a rare event, we therefore use the presence of venulitis as a feature in addition to the presence of autoantibodies and a similar molecular signature to AIP patients as collective parameters to study the nature of the disease.

Letter figure 3. Immunohistochemical stainings of markers of infiltrating cells on pancreatic tissue from a mouse developing spontaneous inflammation after 18 weeks of tamoxifen food treatment. Scale bar: 200 μ m.

5. Mechanistic studies of NEMO-deleted PSC are insufficient. Authors should confirm cell proliferation, migration and ECM production in NEMO-deleted PSC. Effects of PSC-conditioned medium on immune cells must be validated, in combination with neutralizing antibodies for specific factors.

Answer to reviewer #1 point 5:

We thank reviewer #1 for these insightful suggestions. To determine their proliferation, we performed PSC isolations from animals treated with tamoxifen and caerulein with the 3-week protocol. We cultured the isolated PSCs for 48 hours to determine their proliferation. The doubling time of WT and NEMO-deleted primary PSCs were 39.19 and 51.84 hours respectively. In parallel, to further verify the contribution of NEMO-deleted PSCs in recruiting immune cells, we performed a transwell migration assay using conditioned media from isolated PSCs. Using a recombinant CCL24 as a control, we found that the conditioned media from NEMO-deleted PSCs induced much stronger chemotactic effects i.e. more transmigrated immune cells. The majority of the migrated cells were SiglecF+, indicating these were eosinophils. This migration was strongly inhibited after the addition of a CCL24 neutralizing antibody to the conditioned media derived from NEMO-deleted PSCs. Therefore, these results suggest that the effect of NEMO-deleted PSCs in recruiting eosinophils was strongly dependent on CCL24. These new results are now included in Fig 8 h-j and main text line 348 to 357.

To compare the ECM production between WT and NEMO-deleted PSCs, we showed that NEMO-deleted PSCs expressed significantly higher *Col1a1* and *Col3a1* (Fig 8f) levels. We performed additional qPCR to study the production of other ECM components or components involved in ECM modifications e.g. *Fn1*, *Mmp9* and *Timp1*. All these genes were significantly upregulated in NEMO-deleted PSCs (Fig S8). These new results support our observations that NEMO^{ΔCol1a2} mice developed stronger fibrosis.

6. Alteration of intracellular signals in NEMO-deleted PSC need to be validated. Whole tissue lysate includes pancreatic acinar cells that should have altered intracellular signals by inflammation.

Answer to reviewer #1 point 6:

We thank reviewer #1 for the suggestion of validating the intracellular signals in PSCs. If we understood correctly from this comment, there was a confusion about the signaling pathway results presented in Figure 5 f-h. The western blots shown were intended to investigate the altered signaling pathways in the whole pancreatic tissue. Since we found a strong upregulation of IL6 expression in the NEMO^{-/-}Col1a2 pancreata after caerulein-induced pancreatitis, we were interested in the affected pathways including STAT3 and MAPK signalings and indeed we observed a stronger activation of these pathways in the NEMO^{-/-}Col1a2 pancreata. These pathways have been reported to play critical roles in the pathogenesis of pancreatitis. For example, the IL6-STAT3 axis in pancreatic acinar cells was reported to promote pancreatitis-associated lung injury and thus increase lethality (Zhang *et al.*, 2013). Although PSCs were known to be able to produce IL6 (Masamune and Shimosegawa, 2009), we could not exclude that other immune cells like myeloid cells were also involved. Several signaling pathways were implicated in the activation and cellular function of PSCs. These included PPAR- α , AP1, NF- κ B, PI3K-Akt, MAPK, JAK-STAT and Sonic hedgehog signaling (Masamune and Shimosegawa, 2009). Due to the limited materials from the isolated cells and the focus of this study, we could not investigate all these pathways. We do have some preliminary data in our follow-up study to look into other signaling pathways like the Hedgehog signaling and the Notch signaling in the PSC^{-NEMO} cells.

[data redacted]

7. Authors stated that this mouse model exhibited key features of AIP in line 268-269, but I cannot agree with this statement. This model represents eosinophil-related pancreatitis model. Authors should clarify to which disease entity this mouse model can be attributed.

Answer to reviewer #1 point 7:

We regret that reviewer #1 did not agree with this statement. As explained in the answer to point 1, we drew our conclusion based on the discussed 4 observations. However, we are willing to rewrite this part as a summary of our general observations in the mouse model rather than to emphasize the similarity to human disease. We also understand the limitations of mouse models that sometimes they could not recapitulate all features presented in patients.

8. Additional knockout of CCL24 will be the best way to confirm that this cytokine is essential for NEMO deletion-induced pancreatitis.

Answer to reviewer #1 point 8:

We agree with reviewer #1 that a CCL24 knockout mouse line would be the best way to verify our finding that CCL24 is important for the phenotype observed in the NEMO^{-/-}Col1a2 mice. However, these experiments cannot be performed within the time limit of the revision. Firstly, we do not have a CCL24 knockout mouse line available in our mouse facility. Secondly, even if we can import the CCL24 mouse line, it will take us 6-9 months to generate a cohort of target mice. Thirdly, to start planning this experiment involving introducing a new mouse line, we need to provide an animal experiment license. Due to all these limitations, we are not able to provide results from the double knockout mouse line in this revision.

Alternatively, as discussed in the answer to point 5, we have now showed that using a conditioned medium from NEMO-deleted PSCs, we observed increased levels of eosinophil migration in the transwell migration assay. Such effect was abolished after adding the CCL24 antibody to the conditioned medium (Fig 8h-j). Therefore, these results provided new evidence that the recruitment of eosinophils was a result of increased CCL24 production and secretion by NEMO-deleted PSCs.

9. There are several typos such as line 69, there is no space before the brackets. Please check.

Answer to reviewer #1 point 9:

We thank reviewer #1 for pointing out these typos. We have now corrected them.

Reviewer #2 (Remarks to the Author):

This manuscript describes a genetically engineered mouse model for autoimmune pancreatitis by conditionally blocking the activation of NF κ B pathway in mesenchymal cells. In addition, they show that the lack of mesenchymal NF κ B signaling increases the fibrotic response to induction of chronic pancreatitis (CP) by repeated cerulein treatment. As such, this is a very interesting and important contribution to the field of pancreatitis research. My expertise in immunology is rather limited and therefore I will restrict my review on the non-immunologic parts of this manuscript.

Response to the general comment of reviewer #2:

We appreciate reviewer #2 for finding our results very interesting and important contribution to the field of pancreatitis research.

1. The authors should introduce the correct nomenclature (homepage of Jax lab is a good source) of the Cre-driver, which is B6.Cg-Tg(Col1a2-cre/ERT,-ALPP)7Cpd/J and the correct nomenclature of IKK-gamma/Nemo floxed allele: Ikkbg^{tm1M_p}. To call the KO mice NEMO ^{Δ Col1a2} is very misleading, as it implies the lack of Col1a2 in my understanding. I would suggest something like Col1a2-CRE^{ERT};Nemo^{fl}, or just call them cdNemo^{KO} for conditional Nemo knockout.

Answer to reviewer #2 point 1:

We thank reviewer #2 for these suggestions. We agree that a suitable nomenclature would improve the readability of this article. We have considered this issue previously. Originally, we intended to use Col1a2.CreERT tg+ /NEMO fl as a nomenclature for our mouse line. However, after a second consideration, we decided to switch to NEMO^{WT} and NEMO ^{Δ Col1a2} to indicate the control and knockout mice. There are a few reasons leading to such a decision on this simplified nomenclature. Firstly, to maximize the use of mice, as a 3R principle in animal experiment, we used littermates which were a

mixture of Col1a2.CreERT tg- NEMO fl and Col1a2.CreERT tg+ /NEMO wt. Secondly, since NEMO is X-linked, male mice were NEMO fl and female mice were NEMO fl/fl. Thirdly, we also generated reporter mouse lines which carried ROSA26.LSL-RFP (Col1a2.CreERT tg+ /NEMO fl / ROSA26.LSL-RFP and Col1a2.CreERT tg+ /NEMO wt / ROSA26.LSL-RFP). Therefore, it is quite complicate to find a common nomenclature by listing their genetic constructs. The cdNemo^{Ko} option is much more simplified but it does not state the target cells or tissue. We believe the nomenclature of NEMO^{WT} and NEMO^{WT.RFP} nomenclature can better summarize the littermate controls, while NEMO^{ΔCol1a2} can give certain information on the specific target cells. To make it more clear to the readers, we now also include a detailed nomenclature in the material and methods part (line 485 to 489).

2. How have the mice been genotyped? Nemo is X-linked, females in experiments had homozygous or heterozygous deletion?

Answer to reviewer #2 point 2:

We thank reviewer #2 for these questions so we have a chance to clarify them. As Nemo is X-linked, male NEMO^{ΔCol1a2} mice are hemizygous i.e. NEMO fl while female NEMO^{ΔCol1a2} mice are homozygous i.e. NEMO fl/fl. We now also updated the description in the material and method part (line 482 to 483).

3. Was there any sex-specific difference in the extent of AIP or CP?

Answer to reviewer #2 point 3:

We thank reviewer #2 for this question. We observed a similar phenotype in both genders. We saw no difference in terms of severity of pancreatitis. We also added this description to the material and methods part (line 483 to 484).

4. Fig. S1b (and S6) shows that recombination has taken place, but not in which cells (double staining?) and to which extent. This important information comes to some extent in Fig. 5c and 8a-d (statistics?). I think it would aid in understanding if the characterisation of the recombination efficiency would be put together in a paragraph early in the manuscript.

Answer to reviewer #2 point 4:

We thank reviewer #2 for these insightful comments. Both in Figure S1b and S6, we observed that the RFP+ cells were present in the interstitial area but not in acinar cells, ductal cells, islets or endothelial cells. We then showed double stainings in Fig 5c, both RFP/CD45 and RFP/aSMA to indicate that only the aSMA+ cells (activated pancreatic stellate cells) were found positive for RFP. We next looked into the recombination efficiency in isolated PSCs. All purified cells showed nestin (marker of PSCs) expression (Fig 8c). For the analysis based on RFP expression, we found out that 56% of isolated PSCs expressed RFP, suggesting the recombination efficiency was 56%. However, when we looked at NEMO expression, we observed a 76% reduction by qPCR (mRNA level) and 92% reduction by Western blot (protein level). Since the labelling of the cells with RFP relies on its expression which is regulated by the promoter at the ROSA26 loci, as well as its cellular half-life, we believe that a direct measurement by qPCR and Western blot on NEMO could be a better estimate for the recombination level. We now also added the protein quantification in Fig 8d. Although we agree with reviewer #2 that the information of recombination efficiency could be better shown earlier in the manuscript, qPCR and Western blot from the whole tissue could not reflect the real recombination efficiency, since all other cells like pancreatic acinar cells and immune cells can express NEMO. We therefore showed the level of NEMO deletion in figure 8d & e when we isolated pure PSCs. We now added a more detailed description on the knockout efficiency together with the characterization of the isolated PSCs (line 327 to 332).

5. What is TOM20?

Answer to reviewer #2 point 5:

We thank reviewer #2 for this question. TOM20 is a mitochondria outer membrane protein. We used it as a loading control as we had seen in many occasions that their level remained relatively unchanged (Chan *et al.*, 2017).

6. Fig. S1a implies analysis 2 weeks after Tam injection, whereas the figure legend reads 3 weeks after Tam injection.

Answer to reviewer #2 point 6:

We thank reviewer #2 for pointing out this issue. We apologize for the confusion in the figure legend. Referring to the main text, what we wanted to state here was 3 weeks after the first tamoxifen injection. We have now clarified in the figure legend.

7. Looks like there are no tables in the downloadable pdf.

Answer to reviewer #2 point 7:

We thank reviewer #2 for pointing this out. We apologize for missing the supplementary tables which should display the list of antibodies for Western blot and FACS, as well as primers for qPCR. We have now added them back to the supplementary information.

8. Line 214: “Fig. 3g” should read “Fig. 4g”.

Answer to reviewer #2 point 8:

We thank reviewer #2 for pointing out this typo. We have now corrected it in the main text.

9. Are AIP patients treated with prednisolone before AIP detectable symptoms? If not, the mice should be treated after TAM/Cer induction to better reflect the clinical situation.

Answer to reviewer #2 point 9:

We thank reviewer #2 for this critical question. The AIP patients are indeed treated with prednisolone when they display symptoms. In our experimental setup, we started the treatment after the first episode of caerulein injections (6 injections). Since repetitive caerulein injections lead to damages to the pancreas and very often the first sign of pancreatitis starts to develop after 6 hours e.g. elevated serum lipase level (Chan *et al.*, 2017), we believe the starting of the prednisolone treatment after 6 hours could be relevant to this model. We now also clarify this in the material and methods part. However, to answer this question, we performed a separate experiment in this revision to start the prednisolone treatment 1 week after the starting of caerulein and tamoxifen.

[data redacted]

References:

Chan, L. K. *et al.* (2017) 'Epithelial NEMO/IKKI 3 limits fibrosis and promotes regeneration during pancreatitis', *Gut*, 66(11), pp. 1995–2007. doi: 10.1136/gutjnl-2015-311028.

Manohar, M. *et al.* (2021) 'Eosinophils in the pathogenesis of pancreatic disorders', *Seminars in Immunopathology*, 43(3), pp. 411–422. doi: 10.1007/s00281-021-00853-0.

Masamune, A. and Shimosegawa, T. (2009) 'Signal transduction in pancreatic stellate cells', *Journal of Gastroenterology*, 44(4), pp. 249–260. doi: 10.1007/s00535-009-0013-2.

Neuhöfer, P. *et al.* (2013) 'Deletion of I κ B activates RelA to reduce acute pancreatitis in mice through up-regulation of Spi2A', *Gastroenterology*, 144(1), pp. 192–201. doi: 10.1053/j.gastro.2012.09.058.

Sah, R. P. *et al.* (2010) 'Eosinophilia and Allergic Disorders in Autoimmune Pancreatitis', 105(11), pp. 2485–2491. doi: 10.1038/ajg.2010.236.

Ulmasov, B. *et al.* (2013) 'Differences in the degree of cerulein-induced chronic pancreatitis in C57BL/6 mouse substrains lead to new insights in identification of potential risk factors in the development of chronic pancreatitis', *American Journal of Pathology*, 183(3), pp. 692–708. doi: 10.1016/j.ajpath.2013.05.020.

Wang, Q. *et al.* (2009) 'Eosinophilia associated with chronic pancreatitis', *Pancreas*, 38(2), pp. 149–153. doi: 10.1097/MPA.0b013e31818d8ecc.

Zhang, H. *et al.* (2013) 'IL-6 trans-signaling promotes pancreatitis-associated lung injury and lethality', *Journal of Clinical Investigation*, 123(3), pp. 1019–1031. doi: 10.1172/JCI64931.

REVIEWERS' COMMENTS:

Reviewer #1 (Remarks to the Author):

In general, authors performed suggested experiments and addressed my questions. Assessment of infiltrating immune cells clarified the cellular population of these cells. Authors histologically evaluated venulitis in detail. Mechanistic studies of NEMO-deleted PSCs revealed increased chemotactic effects on immune cells. Even though preliminary results, altered intracellular signaling pathways in PSCs would be an adequate target for the next study. Inhibitory effects of CCL24 neutralizing antibody suggested certain contribution of this cytokine to the cell-to-cell interaction. Now, this manuscript seems appropriate for publication. Thank you for addressing these points accordingly.

Reviewer #2 (Remarks to the Author):

All questions have been answered to the reviewer's satisfaction. Congratulations!